# Highly resolved mapping of NO₂ vertical column densities from GeoTASO measurements over a megacity and industrial area during the KORUS-AQ campaign

Gyo-Hwang Choo[1], Kyunghwa Lee[1], Hyunkee Hong[1*], Ukkyo Jeong[2,3], Wonei Choi[4], Scott J. Janz[3]

[1]Environmental Satellite Center, National Institute of Environmental Research, Hwangyeong-ro 42, Seo-gu, Incheon, Republic of Korea, 22689

[2]Earth System Science Interdisciplinary Center, University of Maryland, College Park, Maryland, USA 20740

[3]NASA Goddard Space Flight Center, Greenbelt, Maryland, USA, 20771

[4]Division of Earth Environmental System Science, Major of Spatial Information Engineering, Pukyong National University, Busan 48513, South Korea

*Correspondence to*: Hyunkee Hong; Tel: +82 32 560 8437; Fax: +82 32 560 8460; E-mail address: wanju77@korea.kr

**Abstract.** The Korea-United States Air Quality (KORUS-AQ) campaign is a joint study between the United States National Aeronautics and Space Administration (NASA) and the South Korea National Institute of Environmental Research (NIER) to monitor megacity and transboundary air pollution around the Korean Peninsula using airborne and ground-based measurements. Here, tropospheric nitrogen dioxide ($NO_2$) slant column density (SCD) measurements were retrieved from Geostationary Trace and Aerosol Sensor Optimization (GeoTASO) L1B data during the KORUS-AQ campaign (May 2 to June 10, 2016). The retrieved SCDs were converted to tropospheric vertical column densities using the air mass factor (AMF) obtained from a radiative transfer calculation with trace gas profiles and aerosol property inputs simulated with the Community Multiscale Air Quality (CMAQ) model and surface reflectance data obtained from the Moderate Resolution Imaging Spectroradiometer (MODIS). For the first time, we examine highly resolved (250 m × 250 m resolution) tropospheric $NO_2$ over the Seoul and Busan metropolitan regions, and the industrial regions of Anmyeon. We reveal that the maximum $NO_2$ VCDs were $4.94 \times 10^{16}$ and $1.46 \times 10^{17}$ molecules cm$^{-2}$ at 9 AM and 3 PM over Seoul, respectively, $6.86 \times 10^{16}$ and $4.89 \times 10^{16}$ molecules cm$^{-2}$ in the morning and afternoon over Busan, respectively, and $1.64 \times 10^{16}$ molecules cm$^{-2}$ over Anmyeon. The VCDs retrieved from the GeoTASO airborne instrument were correlated with those obtained from the Ozone Monitoring Instrument (OMI) (r = 0.48), NASA's Pandora Spectrometer System (r = 0. 91), and $NO_2$ mixing ratios obtained from in situ measurements (r = 0.07 in the morning, r = 0.26 in the afternoon over the Seoul, and r > 0.56 over Busan). Based on our results, GeoTASO is useful for identifying $NO_2$ hotspots and their spatial distribution in highly populated cities and industrial areas.

## 1 Introduction

Nitrogen dioxide ($NO_2$) is one of the most important atmospheric trace gases and plays a key role in aerosol production and tropospheric ozone photochemistry (Boersma et al., 2004; Richter et al., 2005). Furthermore, high concentrations of $NO_2$ in the atmosphere have adverse effects on human health, such as respiratory infections, and associated symptoms (Brauer et al., 2002; Latza et al., 2009).

The main sources of $NO_2$ in the atmosphere are fossil fuel combustion from vehicles and thermal power plants, lightning, and biogenic soil processes. Furthermore, $NO_2$ concentrations are highly correlated with population size (Lamsal et al., 2013). The implementation of emission control technology and environmental regulation has led to a decrease in surface $NO_2$ concentrations in Western Europe, the United States, and Japan in the last few decades (Richter et al., 2005). The concentration of $NO_2$ in major metropolitan cities in South Korea and China is over 3 times larger than over similarly sized cities in Europe and United States, despite $NO_2$ concentration decreasing in China and South Korea (de Foy et al., 2016, Choo et al., 2020).

To date, several low-orbit space borne sensors, such as the Global Ozone Monitoring Experiment (GOME) (Burrows et al., 1999), the Scanning Imaging Spectrometer for Atmospheric Cartography (SCIAMACHY) (Burrows et al., 1995), the Ozone

Monitoring Instrument (OMI) (Levelt et al., 2006), the GOME-2 (Callies et al., 2000), and the Tropospheric Monitoring
Instrument (TROPOMI) (Veefkind et al., 2012), have monitored atmospheric ozone and its precursors including $NO_2$ and
formaldehyde (HCHO) as a proxy for volatile organic compounds (VOCs). Furthermore, the Geostationary Environment
Monitoring Spectrometer (GEMS) (Choi et al., 2018; Kim et al., 2020), which was launched on February 18, 2020, will form
a constellation of geostationary satellites including the upcoming Tropospheric Emission: Monitoring of Pollution (TEMPO)
(Zoogman et al., 2017) and Sentinel-4 platforms, to continuously observe the air quality of the Northern Hemisphere during
the day.
$NO_2$ retrievals from space borne hyperspectral measurements are typically conducted using the differential optical absorption
spectroscopy (DOAS) method (Platt and Stutz, 2008) to first retrieve the view-dependent slant column density (SCD), and
then radiative transfer models are used to determine the vertical column density (VCD) using an air mass factor (AMF)
correction. Previous and ongoing space borne instruments use various radiative transfer codes and model input assumptions to
calculate $NO_2$ AMF values at coarse spatial resolution. Because AMF weighting has a large impact on $NO_2$ retrievals using
the DOAS method, it is important to use model input assumptions that most accurately match viewing and atmospheric
conditions. Several studies have demonstrated the sensitivity of AMF calculations to inaccurate model input parameters (e.g.,
*a priori* $NO_2$ vertical profile and aerosol properties) and *a priori* data (cloud information and surface reflectance) (Leitão et
al., 2010; Hong et al., 2017; Lorente et al., 2017; Boersma et al., 2018). $NO_2$ retrievals have also been consistently conducted
based on surface remote sensing measurements including the Multi-Axis DOAS (MAX-DOAS), Système D'Analyse par
Observations Zènithales (SAOZ) spectrometer (Pastel et al., 2014), and Pandora (Herman et al., 2009) systems. These ground-
based measurements can be used as validation references for both airborne and space borne measurements.
$NO_2$ retrievals from airborne remote sensing instruments, such as the Geostationary Coast and Air Pollution Event (GEO-
CAPE) Airborne Simulator (GCAS) (Kowalewski and Janz, 2014), the Heidelberg Airborne Imaging DOAS Instrument
(HAIDI) (General et al., 2014), the Geostationary Trace gas and Aerosol Sensor Optimization (GeoTASO) (Leitch et al., 2014),
the Airborne Prism Experiment (APEX; Popp et al., 2012), the Airborne Imaging DOAS instrument for Measurements of
Atmospheric Pollution (AirMAP; Meier et al., 2017; Schönhardt et al., 2015), the Small Whiskbroom Imager for atmospheric
compositioN monitorinG (SWING; Merlaud et al. 2018), and the Spectrolite Breadboard Instrument (SBI; Vlemmix et al.,
2017; Tack et al., 2019) have also been performed to identify local emission sources and obtain highly resolved horizontal
$NO_2$ distributions.
Observations using airborne measurements have an advantage as they enable the observation of horizontal distributions of
trace gases at resolutions higher than those of space-based satellites and provide data over a wider area than those of ground-
based observations. For example, Nowlan et al. (2018) retrieved tropospheric $NO_2$ VCDs over Houston, Texas, during the
Deriving Information on Surface Conditions from Column and Vertically Resolved Observations Relevant to Air Quality
(DISCOVER-AQ) campaign and identified a high correlation with data retrieved from Pandora. Popp et al. (2012) also
presented the morning and afternoon $NO_2$ spatial distribution in Zurich, Switzerland, using APEX. Tack et al. (2017) have
conducted high-resolution mapping of $NO_2$ over three Belgium cities (Antwerp, Brussels, and Liège) using APEX and Judd
et al. (2020) and Tack et al. (2021) compared $NO_2$ VCDs retrieved from GCAS/GeoTASO and APEX with those obtained
from TROPOMI over New York City and Antwerp and Brussels, respectively. Merlaud et al. (2013) observed $NO_2$ VCDs in
Turceni over Romania using SWING mounted on an unmanned aerial vehicle (UAV) during the Airborne Romanian
Measurements of Aerosols and Trace gases (AROMAT) campaign. These existing $NO_2$ retrievals, using airborne
measurements, have been useful in constraining regional air quality models due to the highly resolved source identification
and the ability to tie these results to ground-based observations.
This work focuses on airborne $NO_2$ retrievals from GeoTASO. This instrument was developed by Ball Aerospace to reduce
mission risk for UV-VIS air quality measurements from geostationary orbit for the GEMS and TEMPO missions (Leitch et
al., 2014). The retrieval of $NO_2$, $SO_2$, and HCHO observed from GeoTASO L1B data using DOAS and principal component
analysis (PCA) (Wold et al., 1987) was conducted through the DISCOVER-AQ and KORea-United States Air Quality
(KORUS-AQ) campaigns (Nowlan et al., 2016; Judd et al., 2018; Choi et al., 2020; Chong et al., 2020). The KORUS-AQ
campaign is a joint study between the National Institute of Environmental Research (NIER) and National Aeronautics and
Space Administration (NASA) to monitor megacity air pollution and transboundary pollution, and to prepare for geostationary
satellite (i.e., GEMS, TEMPO, and Sentinel-4) air quality observability (of trace gases and aerosols), organized from May to
June 2016.
Although surface $NO_2$ concentrations in South Korea are the high due to high population density, high traffic volumes, and
many industrial complexes and thermal power plants, and although $NO_2$ retrieval studies using airborne and ground
measurements in North America, Europe, China, and Japan, data for South Korea remain limited. The specific objectives of
this study are as follows:
(1)  To retrieve tropospheric $NO_2$ vertical column data using GeoTASO measurements over polluted regions of the Seoul

95        and Busan metropolitan areas and the Anmyeon industrial regions of the Korean Peninsula.

(2)  To estimate $NO_2$ VCD uncertainties using error propagation accounting for spectral fitting errors and AMF

97        uncertainties associated with input data errors, including aerosol optical depth (AOD), single scattering albedo (SSA),

98        aerosol peak height (APH), and surface reflectance (SRF).

(3)  To compare $NO_2$ VCDs retrieved from GeoTASO and those obtained from OMI and ground-based Pandora

100        instruments, as well as surface in situ measurements.

**2 KORUS-AQ campaign area, measurements, and model simulation**
**2.1 Campaign area**

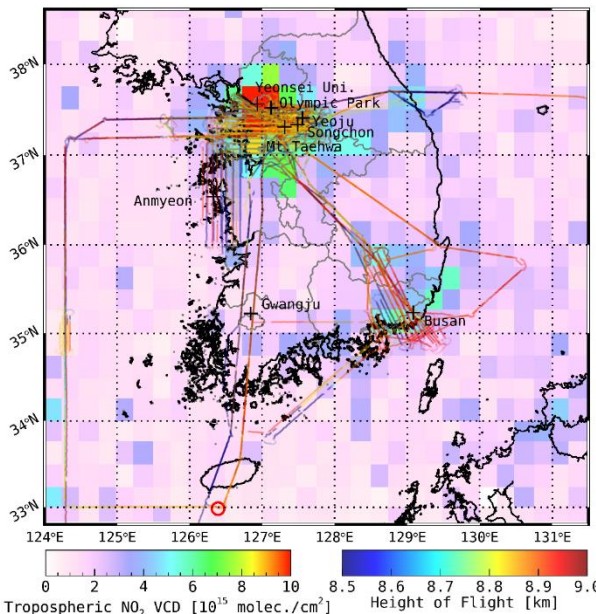

**Figure 1. Flight paths of the NASA LaRC B200 aircraft carrying GeoTASO and the average tropospheric $NO_2$ VCDs**
**obtained from OMI gridded to a $0.25° \times 0.25°$ horizontal grid during the KORUS-AQ campaign period. The line color**
**represents flight height. In this period, the GeoTASO observations focused on megacities (Seoul and Busan) and**
**industrial complex area (Anmyeon) with high tropospheric $NO_2$ concentrations. The reference spectrum for spectral**
**fitting is obtained from the radiation data over the Jeju Island (marked with red circle).**
The Korean Peninsula, located on the Asia-Pacific coast, has a complex atmospheric environment by local emissions and long-
range transport under appropriate weather conditions (Jeong et al., 2017; NIER and NASA, 2020; Choo et al., 2021). Seoul,
the capital of South Korea, and the metropolitan area are densely populated, and power plants and industrial activities on the

northwest coast are conducted, which emit relatively large amounts of pollutants. The KORUS-AQ campaign conducted three-dimensional observations, including ground-based remote, aircraft, satellite observation, and air quality modeling, to understand the complex air quality and interpret the observations of GEMS launched in 2020. The KORUS-AQ campaign period was from May 2 to June 10, 2016. During the KORUS-AQ campaign, air pollutants were conducted using the GeoTASO on board the NASA Langley Research Center B200 aircraft to monitor air quality and long-range transport of pollutants over the Korean Peninsula (NIER and NASA, 2020). The GeoTASO observations were conducted 30 times in 23 d out of 40 d. Most observations were made once or twice a day. Each flight was planned and conducted on a day when the weather conditions were fine and flight hours were approximately 2-4 h. We show the average values of GeoTASO flight information such as flight time, altitude, speed, solar zenith angle (SZA), and viewing zenith angle (VZA) for the dates retrieved for $NO_2$ VCD, aerosol properties (AOD, SSA) extracted from CMAQ, and cloud fraction and surface reflectance extracted from the Moderate Resolution Imaging Spectroradiometer (MODIS) in Table 1. Flight information on the date of aircraft observation can be found at http://www-air.larc.nasa.gov/missions/korus-aq/docs/KORUS-AQ_Flight_Summaries_ID122.pdf. Figure 1 indicates the flight routes of B200 and the tropospheric $NO_2$ VCD obtained from the OMI during the campaign period. The observations were concentrated in the metropolitan areas of Seoul and Busan and the industrial areas of Anmyeon, with an average flight altitude of ~8.5 km during KORUS-AQ.

Table 1. Summary of information on the dates when $NO_2$ VCD was retrieved during the KORUS-AQ period (LT = UTC + 9 h). The average values of GeoTASO data sets for flight characteristics, aerosol properties, geometric information and cloud information.

| Date | Jun 5 | Jun 9 AM | Jun 9 PM | Jun 10 AM | Jun 10 PM |
|---|---|---|---|---|---|
| ROI | Anmyeon | Seoul metropolitan | | Busan metropolitan | |
| Flight time (LT) | 13:11–17:20 | 7:48–12:00 | 13:46–17:52 | 8:02–11:38 | 13:05–15:19 |
| Flight altitude (km) | 8.6 | 8.4 | 8.5 | 8.6 | 8.5 |
| Flight speed (km hr$^{-1}$) | 117.0 | 116.2 | 117.6 | 117.2 | 117.1 |
| SZA (°) | 39.2 | 36.1 | 45.3 | 35.9 | 33.0 |
| VZA (°) | 11.9 | 12.6 | 12.8 | 12.1 | 11.8 |
| AOD | 0.27 | 0.40 | 0.21 | 0.13 | 0.09 |
| SSA | 0.966 | 0.980 | 0.949 | 0.981 | 0.968 |
| Surface reflectance | 0.07 | 0.09 | 0.09 | 0.06 | 0.06 |

| | | | | | |
|---|---|---|---|---|---|
| Cloud fraction | 0.08 | 0.31 | 0.55 | 0.16 | 0.20 |


As shown in Fig. 1, GeoTASO observations were conducted focusing on highly $NO_2$-polluted regions in the Seoul and Busan
metropolitan areas and the Anmyeon region during the KORUS-AQ campaign. The Seoul metropolitan area (Seoul Special
City, Gyeonggi Province, and Incheon City) is one of the most densely populated areas worldwide, with a population of
approximately 20 million in 2016. Busan is the second-largest city in South Korea, with a population of approximately 3.4
million in 2016. Anmyeon is located southwest of Seoul, with petrochemical complexes, steel mills, and thermal power stations
in this area. The background color in Fig. 1 represents the average $NO_2$ VCD obtained from the OMI during the KORUS-AQ
campaign period, showing over $1 \times 10^{16}$ molecules $cm^{-2}$ over the Seoul metropolitan area. The OMI data were obtained with
the Level 2.0 OMNO2 version 3.0 and downloaded from the NASA Earthdata search (http://search.earthdata.nasa.gov/search/).
We calculated the arithmetic means of tropospheric $NO_2$ VCDs, like Choo et al. (2020), to obtain the grid data (0.25° × 0.25°)
during the KORUS-AQ period. The average tropospheric $NO_2$ VCD data were excluded from May 30 2016 to Jun 9 2016,
when the OMI data did not exist during the campaign period.
**2.2 Pandora**
$NO_2$ VCDs retrieved from the GeoTASO were validated using those from NASA's Pandora Spectrometer system. The Pandora
spectrometer is a hyper-spectrometer that can provide direct sun measurements of UV-Vis spectra (280–525 nm with a full
width at half maximum (FWHM) of 0.6 nm) for observing atmospheric trace gases. During the KORUS-AQ, eight Pandora
instruments monitored $NO_2$ and ozone ($O_3$) VCD as depicted by plus symbols in Fig. 1. The retrieved data are available on the
KORUS-AQ pages of NASA's Goddard Space Flight Center website
(https://avdc.gsfc.nasa.gov/pub/DSCOVR/Pandora/DATA/KORUS-AQ/). We compared $NO_2$ VCDs obtained from five
Pandora measurements (Busan university: 35.24 °N, 129.08 °E; Olympic park: 37.52 °N, 127.13 °E: Songchon: 37.41 °N,
127.56 °E; Yeoju: 37.34 °N, 127.49 °E; Yonsei University: 37.56 °N, 126.93 °E) within 0.05° and 30 min with those from
GeoTASO. Because $NO_2$ has a short atmospheric lifetime, especially during the summer (Shah et al., 2020), its spatial and
temporal distributions vary notably. A detailed description of Pandora's operation during the KORUS-AQ campaign has
previously been reported (Herman et al., 2018; Spinei et al., 2018).
**2.3 Ground-based in situ $NO_2$ measurement**
Although the basic physical quantity of VCD and the surface mixing ratio from in situ measurements are different, comparison
of their spatiotemporal variations provides useful information for deriving surface air quality from airborne instruments (e.g.,
Jeong and Hong, 2021a; 2021b). In this study, we compare the $NO_2$ VCDs (molecules $cm^{-2}$) retrieved from GeoTASO to
surface mixing ratios measured by ground-based in-situ monitoring network over South Korea (i.e., Air-Korea, a national real-
time air quality network; https://www.airkorea.or.kr/). The instruments use the chemiluminescence method (Kley and
McFarland, 1980), and approximately 400 air quality monitoring sites in Korea are registered in the system, providing hourly
surface $NO_2$ concentrations. We compared $NO_2$ VCDs retrieved from GeoTASO within 0.5 km and 30 min with $NO_2$
concentrations obtained from Air-Korea.
**2.4 GeoTASO measurements**
$NO_2$ VCDs were retrieved from the L1B radiance dataset (version: V02y) obtained using GeoTASO during the KORUS-AQ
campaign. The NASA Goddard Space Flight Center conducted the L1B radiance calibration, which included offset and smear
correction, gain matching, amplifier cross-talk correction, dark rate correction, integration normalization, sensitivity derivation,
wavelength registration, geo-registration, non-linearity correction, and ground pixel geolocation (Kowalewski et al., 2017;
Chong et al., 2020). The detailed specifications of GeoTASO are listed in Table 2 (Nowlan et al., 2016).

**Table 2. Summary of the GeoTASO instrument and optical specification.**

| L1B version | V02y |
|---|---|
| **Full cross-track field of view** | 45° |
| **Single-pixel cross-track field of view** | 0.046° |
| **Wavelength** | UV: 290–400 nm |
| | VIS: 415–695 nm |
| **Spectral resolution** | UV: ~0.39 nm |
| **(full width at half maximum, FWHM)** | VIS: ~0.88 nm |
| **CCD** | 1,056 (wavelength) × 1,033 (cross-track) |
| **Spatial resolution before binning** | ~35 m (along-track) × 7 m (cross-track) |
| **Spatial resolution after binning** | ~250 m (along-track) × 250 m (cross-track) |


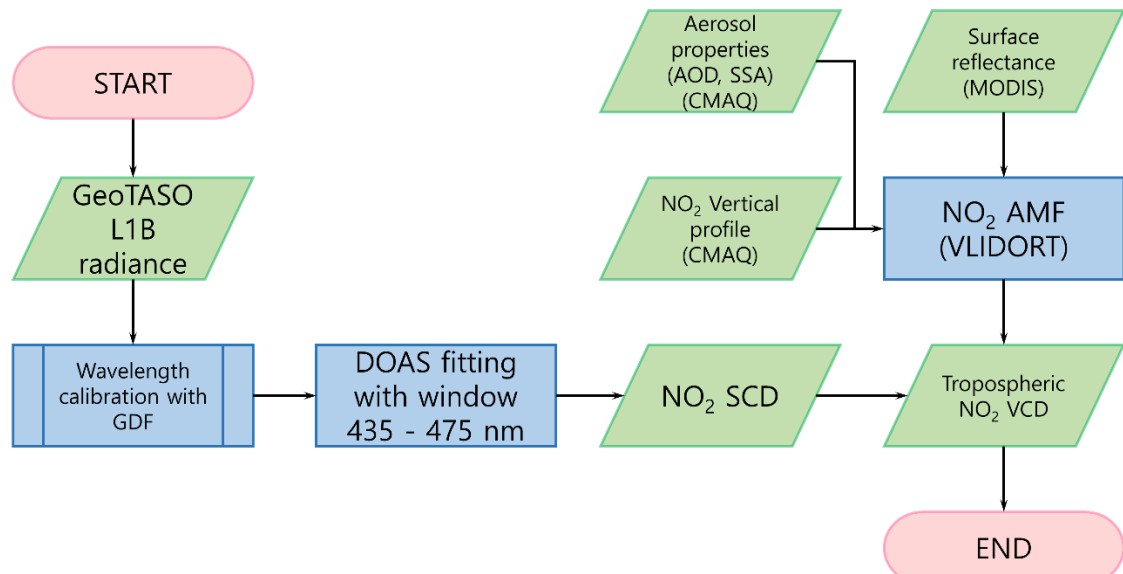


**Figure 2. Flowchart of the algorithm for retrieving tropospheric NO₂ data from GeoTASO.**

**2.4.1 NO₂ slant column density retrieval**
Figure 2 indicates the flowchart for retrieving the tropospheric $NO_2$ VCD from the GeoTASO. We first retrieved $NO_2$ SCDs
using the DOAS method (Platt, 1994). Nonlinear least square minimization was used to retrieve the $NO_2$ SCDs which
minimizes the difference between the measured optical depth and the modeled value in QDOAS software (Eq. (1); Danckaert
et al., 2012).
$$\frac{\ln I(\lambda)}{\ln I_0(\lambda)} = -\left(\sum_{j=1}^{m} \rho_j \times \sigma'_j(\lambda) + B(\lambda) + R(\lambda) + A(\lambda) + N(\lambda)\right),$$ (1)

Where $I(\lambda)$ is the measured earthshine radiance at wavelength $\lambda$; $I_0$ is the reference radiance from the reference sector (southern
ocean of the Jeju Island denoted as the red circle in Fig. 1; 32.983°N, 126.392°E) at 9 AM on May 1 2016. The Community
Multiscale Air Quality (CMAQ) modeling system data indicated that the $NO_2$ VCD from the surface to 50 hPa over this
reference sector on this day was $6.75 \times 10^{15}$ molecules cm-2, and the mean of total $NO_2$ VCD obtained from the OMI during
the KOURS-AQ period was $4.77 \times 10^{15}$ molecules cm-2 with a standard deviation of $1.33 \times 10^{15}$ molecules cm-2. We also
confirmed the stability of $NO_2$ distribution over this area using the TROPOMI offline data from 2019 to 2020. In this period,
the $NO_2$ VCD from the TROPOMI was $4.81 \times 10^{15}$ molecules cm-2 with a standard deviation of $0.43 \times 10^{15}$ molecules cm-2.
The $NO_2$ VCD used as a reference sector obtained from CMAQ was mainly dominated by stratospheric $NO_2$ VCD. However,
stratospheric $NO_2$ VCD has a relatively lower than tropospheric $NO_2$ VCD. The $\rho_j$ represents the SCD of each species $j$; $\sigma'_j(\lambda)$
represents the differential gas phase absorption cross-section convolved with the Gaussian distribution function (GDF) with
GeoTASO FWHM (the UV and VIS range were 0.34−0.49 nm and 0.70−1.00 nm, respectively (Nowlan et al., 2016)) at
wavelength $\lambda$ of species $j$, respectively.
We used the measured radiances at the reference sector to calculate differential slant column density (dSCD) over the whole
domain of the GetoTASO measurements. CMAQ calculation over the reference sector (i.e., $6.75 \times 10^{15}$ molecules cm-2) was
adopted as the reference SCD ($SC_0$), which is added to all dSCD values to convert to the SCD. The reference sector is known
as a background area but is occasionally affected by the long-range transport of $NO_2$ from upwind areas. Considering the
standard deviation of the OMI measurements accounts for such effects during the measurement period, we estimate the
maximum uncertainties of the $SC_0$ can be calculated from this value (i.e., $1.33 \times 10^{15}$ molecules cm-2) in addition to the
difference of the mean values between the CMAQ and OMI (i.e., $1.98 \times 10^{15}$ molecules cm-2). Therefore, our best estimate of
the uncertainty of the $SC_0$ is the root of the sum of squares of these values (i.e., $2.38 \times 10^{15}$ molecules cm-2).

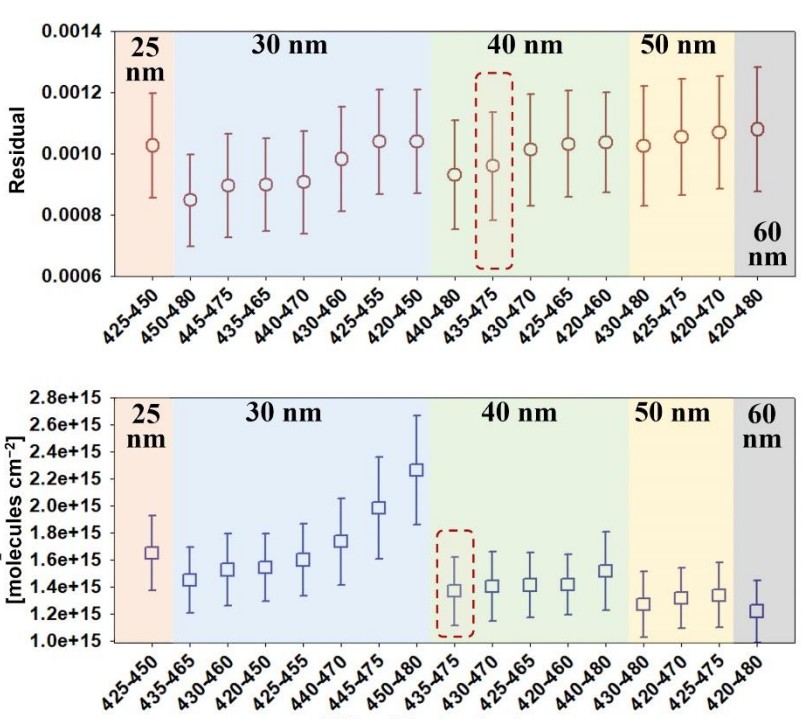


**Figure 3. Residuals and NO₂ SCD errors of 17 spectral fitting window candidates (May 17, 2016, across track number: 15).**

The spectral fitting window was selected based on the sensitivity test with 17 fitting window candidates from 420 to 480 nm
with the length of the fitting window from 25 to 60 nm. Spectral fitting residuals and $NO_2$ SCD errors have been investigated
for 17 spectral fitting window candidates (Fig. 3).
In terms of the residual, when the $NO_2$ fitting window includes a wavelength region less than 430 nm, it has a larger residual
compared to the case where it does not. The higher residual can include the more noise signals that cannot be calculated
mathematically, which can become an uncertainty for the $NO_2$ SCD retrievals. Therefore, we excluded the fitting window
which includes wavelengths less than 430 nm for the GeoTASO $NO_2$ retrievals during the KORUS-AQ campaign. In the case
of the $NO_2$ SCD error, it was confirmed that the longer the fitting window length, the lower the $NO_2$ SCD error appeared
regardless of including the wavelength region less than 430 nm. Therefore, for the stable $NO_2$ SCD retrieval, an appropriate
spectral fitting window needs to be selected which can minimize the residual with a moderate length of the fitting window. To
find the optimal fitting window, we set the threshold value based on the above results: residual < 0.001, $NO_2$ SCD error < 1.4
$\times 10^{15}$ molecules $cm^{-2}$, the length of fitting window > 30 nm. Then, the fitting window of 435–475 nm was selected for the
GeoTASO $NO_2$ retrievals during the KORUS-AQ campaign. To determine the wavelength registration more accurately in the
narrow fitting window, additional wavelength calibration of the spectra for each of the 33 across track pixels was performed
using a high-resolution solar reference spectrum (Kurucz solar spectrum) (Chance and Kurucz, 2010) with the GDF. The
absorption cross-sections of $NO_2$ (Vandaele et al., 1998), $O_3$ (Bogumil et al., 2000), $H_2O$ (Rothman et al., 2010), and the Ring
effect as pseudo-absorbers (Chance and Spurr, 1997) were used to construct the model equation; and $B(\lambda)$, $R(\lambda)$, $A(\lambda)$, and
$N(\lambda)$ are the broad absorption of trace gases, extinction by Mie and Rayleigh scattering, variation in the spectral sensitivity of
the detector or spectrograph, and noise, respectively, which were accounted for by an 8th order polynomial. An example of the
spectral fitting results is presented in Fig. 4.

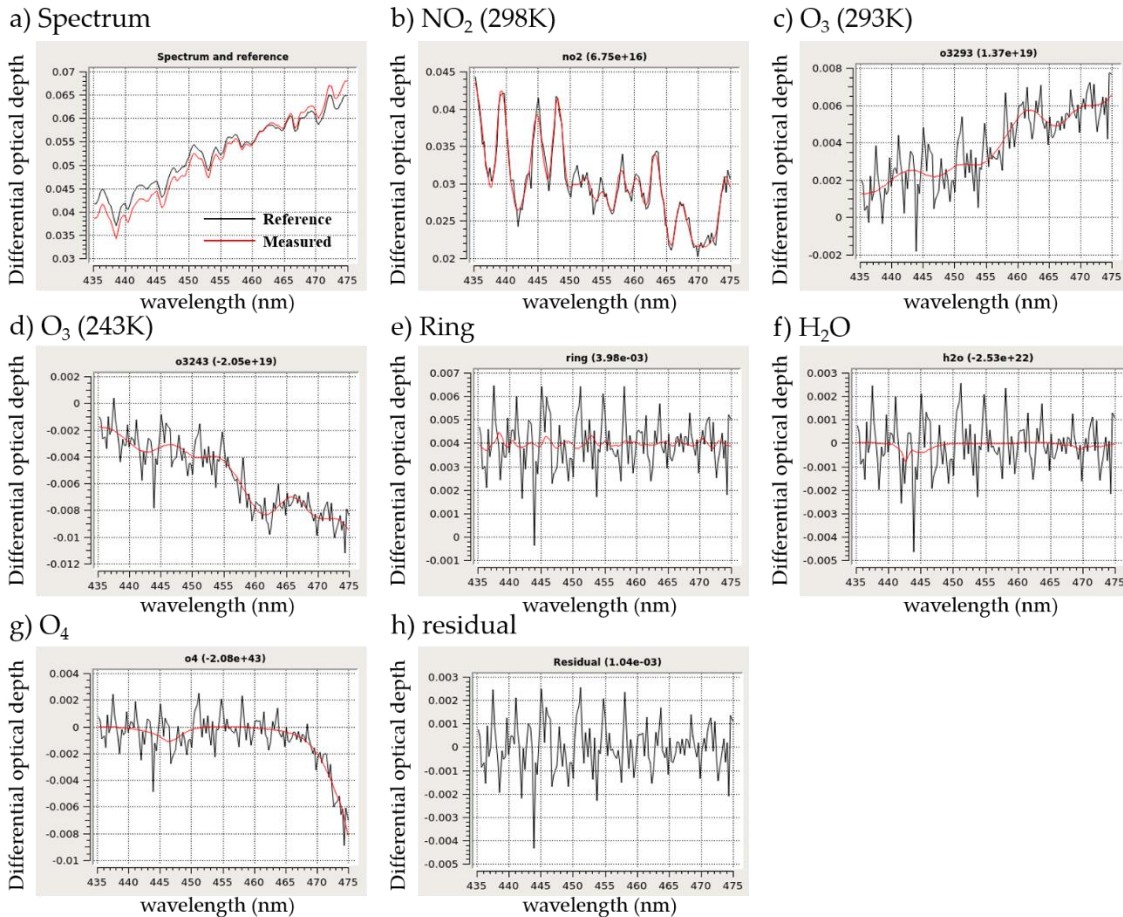


**Figure 4. An example of the spectral fitting results of $NO_2$ retrievals from GeoTASO during the KORUS-AQ campaign (at Gangnam,**
**Seoul on June 9, 2016). Red and black line in the panel (a) represent measured and reference spectrum, respectively. The panels of**
**(b) to (h) depict examples of spectral fitting results of (b) $NO_2$, (c) $O_3$ (293 K), (d) $O_3$ (243 K), (e) ring, (f) $H_2O$, (g) $O_4$, where red and**
**black lines are the absorption cross section of target species and the fitting residual plus the absorption of the target species,**
**respectively. The panel (h) indicates the fitting residual of this example.**

### 2.4.2 NO₂ AMF calculation

AMF, the ratio of SCD to VCD, can be calculated using the scattering weight ($\omega$) and shape factor (S) (Palmer et al., 2001) in Eq. (2)−(5).

$$AMF = \frac{SCD}{VCD},\tag{2}$$

$$AMF = AMF_G \int_{z1}^{z2} \omega(z)S(z)dz,\tag{3}$$

$$\omega(z) = -\frac{1}{AMF_G}\frac{\partial \ln I_B}{\partial \tau},\tag{4}$$

$$S(z) = \frac{\alpha(z)n(z)}{\int_{z1}^{z2}\alpha(z)n(z)dz},\tag{5}$$

Where $AMF_G$ represents the geometric AMF, $I_B$ is the earthshine radiance, $\tau$ is the optical depth, $\alpha$ is the absorption cross-section, and $n$ is the number density of the absorber. NO₂ AMF was calculated using a linearized pseudo-spherical scalar and vector discrete ordinate radiative transfer model (VLIDORT, version 2.6; Spurr and Christi, 2014). Aerosol properties, such as AOD, SSA, APH, and *a priori* NO₂ vertical profile information, were simulated using the CMAQ, and surface reflectivity was obtained from MODIS (Collection 6). The surface reflectance products, MCD43A3, available at a 500 m spatial resolution, provide an estimate of the surface spectral reflectance including MODIS bands 1 through 7. Here, MODIS band 3 (459–479 nm) was used, because this band is the closest the wavelength (455 nm) used in the calculation of AMF in this study. APH was assumed to be the peak height of the aerosol extinction coefficient simulated in CMAQ, and the aerosol profile applied GDF based on APH (Hong et al., 2017). For pixels without reflectance information, AMF was not calculated. The products were corrected for atmospheric conditions, such as aerosol, gases, and Rayleigh scattering. In previous studies (Lamsal et al., 2017; Nowlan et al., 2018; Judd et al., 2019; Chong et al., 2020), an AMF was described for both above and below aircraft altitude is used to convert NO₂ SCDs to VCDs using Eq. (6)−(8).

$$AMF\uparrow = AMF_G \int_{Z_A}^{Z_{TOA}} \omega(z)S(z)dz,\tag{6}$$

$$AMF\downarrow = AMF_G \int_{Z_0}^{Z_A} \omega(z)S(z)dz,\tag{7}$$

$$NO_2\,VCD\downarrow = \frac{NO_2\,SCD - AMF\uparrow \cdot NO_2 VCD\uparrow}{AMF\downarrow},\tag{8}$$

Where $AMF\uparrow$ and $AMF\downarrow$ are AMF above and below aircraft, respectively, and $NO_2\,VCD\uparrow$ represents NO₂ VCD above the aircraft obtained from a chemical transport model (CTM). However, here we calculated NO₂ VCD↓ by dividing NO₂ SCDs by $AMF\downarrow$ as the CMAQ only simulates the troposphere (surface to 50 hPa). However, as the stratospheric and free tropospheric NO₂ ($NO_2\,VCD\uparrow$) column densities over megacities and industrial areas are much lower than tropospheric NO₂ column densities, (Valks et al., 2011), we assume that the uncertainties in the *AMF* without considering the upper atmosphere are negligible in this study.

### 2.5 Chemical model description

Vertical profiles from CMAQ (Byun and Ching, 1999; Byun and Schere, 2006), a CTM, were used to calculate AMFs. The CMAQ simulations were conducted with a horizontal resolution of $15 \times 15$ km and had 27 vertical layers from the surface to 50 hPa. The meteorological fields were prepared using the advanced research Weather Research and Forecasting (WRF) Advanced Research WRF (ARW) Model (Skamarock et al., 2008). Anthropogenic emissions were generated based on the KORUS v5.0 model (Woo et al., 2012), and biogenic emissions were simulated using the Model of Emissions of Gases and Aerosols from Nature (MEGAN v2.1; Guenther et al., 2006; 2012). Besides anthropogenic and biogenic emissions, the Fire Inventory from NCAR (FINN; Wiedinmyer et al., 2006, 2011) was used to update the pyrogenic emission fields.

The CMAQ AOD was calculated by integrating the aerosol extinction coefficient ($Q_{ext}$), which is the sum of scattering ($Q_{sca}$)
and absorption ($Q_{abs}$) coefficients, over all vertical layers ($z$) as follows:
$AOD = \int Q_{ext}(z)\,dz = \int \{Q_{sca}(z) + Q_{abs}(z)\}\,dz,$ (9)
$Q_{abs}[Mm^{-1}] = \sum_i \sum_j \{(1 - \omega_{ij}) \cdot \beta_{ij} \cdot f_{ij}(RH) \cdot [C]_{ij}\},$ (10)
$Q_{sca}[Mm^{-1}] = \sum_i \sum_j \{\omega_{ij} \cdot \beta_{ij} \cdot f_{ij}(RH) \cdot [C]_{ij}\},$ (11)
Here, $\omega_{ij}$ indicates SSA of particulate species i for the particulate mode (or size bin) j, $\beta_{ij}$ denotes the mass extinction
efficiency, $f_{ij}(RH)$ is the hygroscopicity factor according to the relative humidity (RH), and $[C]_{ij}$ is the concentration of
particulate species. CMAQ SSA is defined as the ratio of the integrated $Q_{sca}$ to AOD, and NO₂ vertical profiles were obtained
from NO₂ concentrations at each vertical layers by conducting CMAQ simulations. Details of the model descriptions and
calculations of optical properties are given by Lee et al. (2020) and Malm and Hand (2007).
**3 Results and discussion**
**3.1 NO₂ VCD retrieval**
**3.1.1 Seoul metropolitan region**
We show the final NO₂ VCDs from 250 m spatial resolution. Because of NO₂ VCD, we selected the dates observed in both
the morning and afternoon during the KORUS-AQ period over the Seoul metropolitan area, Busan, and Anmyeon. The
retrieved dates for NO₂ VCDs were Jun 5, 9, and 10, 2016.
The population of the Seoul metropolitan region is approximately 20 million, which is approximately 40% of the total
population of South Korea. It is rare to obtain high-resolution horizontal NO₂ VCD distributions using airborne measurements
in the morning and afternoon, especially in Asian megacities. Figure 5 indicates tropospheric NO₂ VCDs over Seoul on June
9 2016, at 9 AM and 3 PM local time (LT). Because of an issue with imaging systems, enlarged views (Fig. 5-Fig. 8) present
a slightly stripy appearance from the GeoTASO observation (Nowlan et al., 2016; Chong et al., 2020).

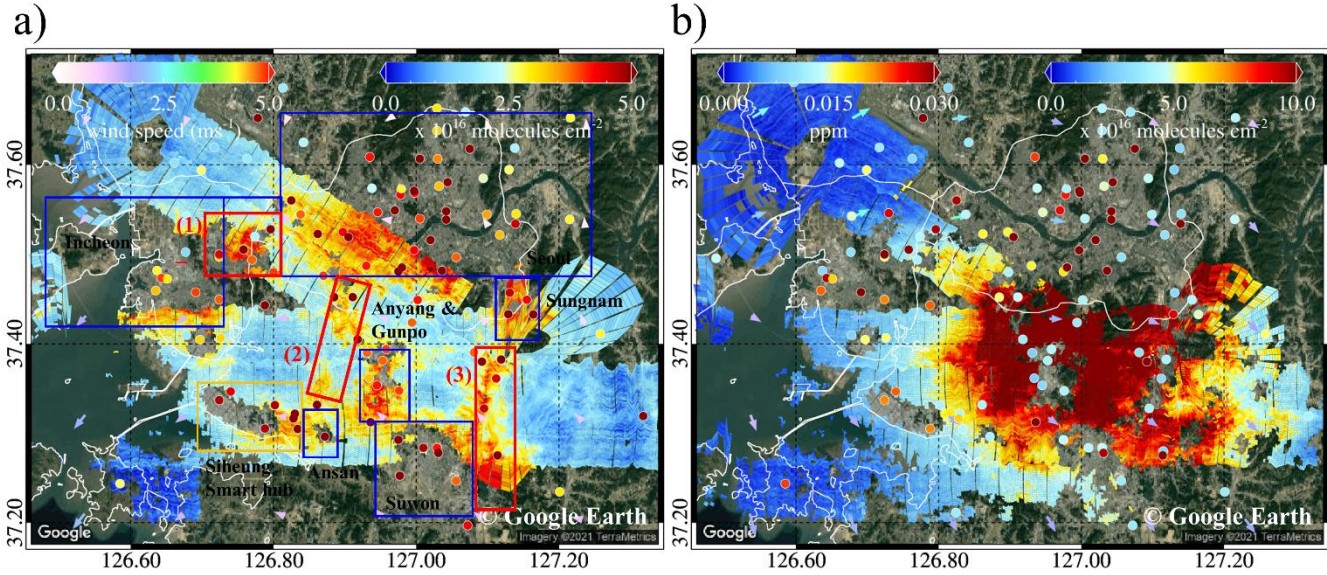

**Figure 5. Tropospheric NO₂ VCD, in the Seoul metropolitan region onJune 9, 2016, retrieved from GeoTASO: a) at 9 AM and b) at**
**3 PM. The red boxes represent expressways (counterclockwise from left to right, (1) Gyeongin Expressway, (2) Seohaean Expressway,**
**and (3) Gyeongbu Expressway), the orange box indicates the industrial complex, and the blue boxes indicate the major cities (Seoul,**
**Incheon, Suwon, Bucheon, Anyang, Gunpo, Sungnam, and Ansan) of the Seoul metropolitan region. Colors of the circles depict the**
**NO₂ surface mixing ratio obtained from Air-Korea. The color arrows indicate the wind direction and speed at 1000 hPa over Seoul**
**metropolitan region, obtained via the Unified Model (UM) simulations (background RGB image is from Google Earth;**
**https://www.google.com/maps/).**

In the morning, NO$_2$ VCDs retrieved from GeoTASO were highly correlated with expressways (red boxes in Fig. 5), such as
the Gyeongin, Seohaean, and Gyeongbu Expressways, and over major cities with heavy traffic, such as Seoul, Bucheon, Ansan,
Anyang, and Suwon. GeoTASO observed NO$_2$ VCD values three-times higher ($>3 \times 10^{16}$ molecules cm$^{-2}$) in these areas
compared to the surrounding rural areas. High NO$_2$ VCD values above $6 \times 10^{16}$ molecules cm$^{-2}$ were observed above the
Gyeongin Expressway, which has very heavy traffic in a relatively short section, and the Gunpo Complex Logistics zone,
where diesel vehicle traffic is also high. The main NO$_2$ source regions and the regions where high NO$_2$ VCD values were
observed were highly consistent at 9 AM because the wind speed at this time—as obtained from the unified model (UM) based
Regional Data Assimilation and Prediction System (RDAPS) of the Korea Meteorological Administration (KMA)—was as
low as 0.1 ms$^{-1}$ and the average wind direction was 84.7° at 1000 hPa over Seoul metropolitan region. The average daily traffic
volume of these expressways exceeds 150,000 vehicles, and the total number of vehicles registered in these major cities is >
6,000,000, with an average daily mileage per car per day of over 38 km. Detailed information on these cities and expressways
is listed in Table 3 and 4. Based on the level of vehicular traffic, combustion using gasoline and diesel engines leads to high
overall emissions of NO$_2$ in the Seoul metropolitan region (Kendrick et al., 2015).

**Table 3. The population, number of registered vehicles, and average mileage per car per day of the major cities in the Seoul**
**and Busan metropolitan region obtained from the Korean Statistical Information Service (https://kosis.kr/eng).**

| City | Population (millions) | Vehicle registration number (thousands) | Average mileage (km car$^{-1}$ day$^{-1}$) |
|---|---|---|---|
| Seoul | 9.776 | 3,083 | 37.1 |
| Incheon | 2.914 | 1,402 | 41.7 |
| Bucheon | 0.848 | 284 | 37.2 |
| Ansan | 0.744 | 289 | 40.8 |
| Anyang | 0.596 | 206 | 39.6 |
| Gunpo | 0.286 | 87 | 38.8 |
| Suwon | 1.241 | 467 | 38.1 |
| Sungnam | 0.994 | 358 | 36.3 |
| Busan | 3.389 | 1,295 | 40.1 |
| Daegu | 2.450 | 1,121 | 37.1 |
| Changwon | 1.080 | 551 | 37.5 |
| Kimhae | 0.529 | 250 | 38.0 |


**Table 4. Daily average traffic volume on the Gyeongin, Gyeongbu, and Seohaean Expressways obtained using the Traffic**
**Monitoring System (https://www.road.re.kr).**

| Expressway | Daily average traffic volume |
|---|---|
| Gyeongin Expressway | 162,369 |
| Gyeongbu Expressway | 173,413 |
| Seohaean Expressway | 150,298 |


Compared to the data of the morning, the average wind speed and wind direction were 1.7 ms$^{-1}$ and 284.5° at 1000 hPa in the
afternoon and the afternoon had extremely high tropospheric NO$_2$ VCD values (exceeding $5 \times 10^{16}$ molecules cm$^{-2}$) in most of
the Seoul metropolitan regions including rural areas, whereas the NO$_2$ mixing ratio (MR) obtained from Air-Korea decreases
in the afternoon. According to Tzortziou et al. (2018), similar results were retrieved from the Pandora site in Seoul, with higher
afternoon NO$_2$ VCDs than in the morning. This result is because the amount of NO$_2$ produced by chemical conversion of nitric
oxide (NO) by O$_3$ and VOCs in the atmosphere, along with NOx generated by regional emissions (traffic) in the Seoul
metropolitan region, is greater than the amount lost by photolysis and transport to nearby areas (Herman et al., 2018).
Furthermore, the increase in tropospheric NO$_2$ VCD in the afternoon is likely due to the accumulation and dispersion of NO$_2$
according to the height of the change in the planetary boundary layer (Ma et al., 2013).
**3.1.2 Industrial and power plant regions in Anmyeon**

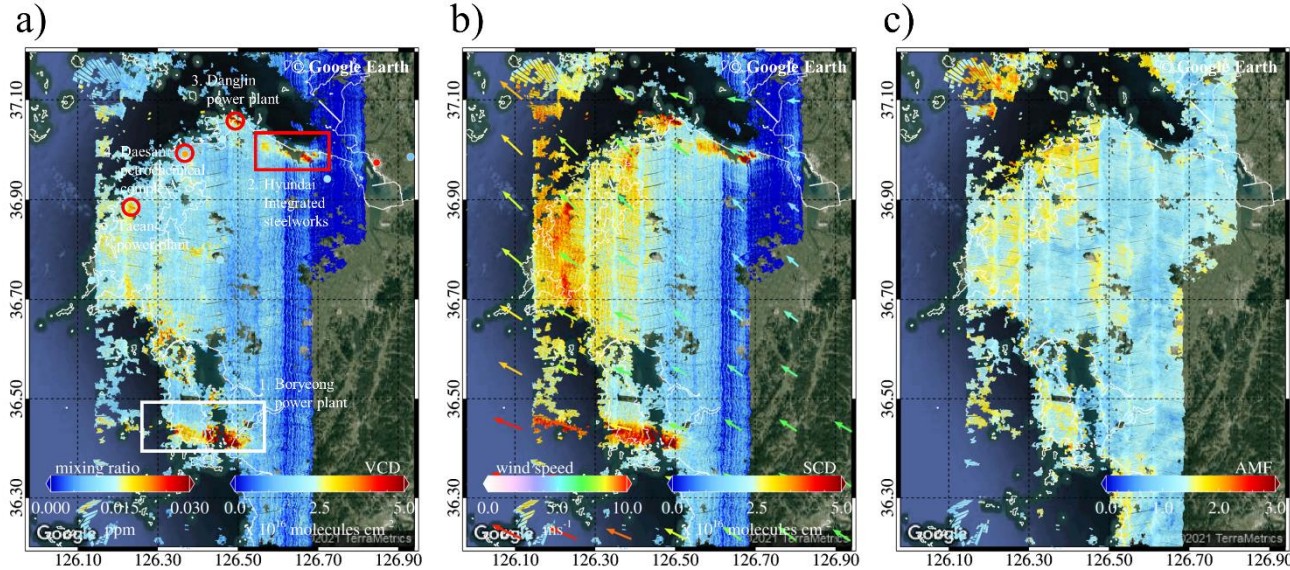


**Figure 6. a) Tropospheric NO$_2$ VCD and b) NO$_2$ SCD retrieved from GeoTASO, and c) NO$_2$ AMF, native resolution (250 m)**
**calculated using VLIDORT over Anmyeon in South Korea on 5 June 2016. The colored arrows indicate wind speed and wind**
**direction at 850 hPa from the Unified Model (UM) simulations. The red circles and rectangle in panel (a) represent the major NO$_2$**
**emission sources, such as steelworks and power plants (background RGB image is from Google Earth;**
**https://www.google.com/maps/).**

The high spatial resolution of the tropospheric NO$_2$ VCD from GeoTASO over the Anmyeon industrial region, where many
industrial facilities and several power plants are distributed, is shown in Fig. 6. Panels a and b of this figure indicate the binned
tropospheric NO$_2$ VCD and NO$_2$ SCD retrieved from GeoTASO L1B data, respectively, between 13:00 and 17:00 LT on June
5 2016. Panel c depicts the calculated AMF of NO$_2$ from native resolution over the domain. GeoTASO observations detected
moderate and strong NO$_2$ emission sources in this area: (1) Boryeong power plant, (2) Hyundai integrated steelworks, (3)
Dangjin power plant, (4) Daesan Petrochemical Complex, and (5) Taean Power Plant. High NO$_2$ VCD values ($> 5 \times 10^{16}$
molecules cm$^{-2}$) were observed over steel mill works, petrochemical complexes, and power plants, whereas values were
comparatively low ($<1 \times 10^{16}$ molecules cm$^{-2}$) over small cities including Seosan, Dangjin, and Boryeong with populations of
less than 0.1 million, and the Seohaean Expressway. In 2016, the annual NOx emissions from Hyundai steelworks and the
Dangjin and Boryeong power plants were approximately 10.3, 11.9, and 16.8 kt year$^{-1}$, respectively. The NOx emission rates
of major industrial facilities in the Anmyeon region are shown in Table 5.

**Table 5. NOx emission rates in 2016 from major industrial facilities in the Anmyeon region obtained from the Continuous**
**Emission Monitoring System of the Korea Environment Corporation (https://www.stacknsky.or.kr/eng/index.html).**

| Industrial facilities | NOx emission rate (kg year$^{-1}$) |
| --- | --- |
| Boryeong power plant | 16,788,438 |
| Hyundai integrated steelworks | 10,271,075 |
| Dangjin power plant | 11,852,972 |
| Daesan petrochemical complex | 3,397,939 |
| Taean power plant | 15,466,022 |


| Industrial facilities | NOx emission rate (kg year$^{-1}$) |
| --- | --- |
| Boryeong power plant | 16,788,438 |
| Hyundai integrated steelworks | 10,271,075 |

Figure 6 shows high $NO_2$ concentrations of the main industrial facilities in the Anmyeon region, where the combustion of
fossil fuel in factories and thermal power plants leads to high emissions (Prasad et al., 2012). Due to relatively sparse
distribution over rural areas, the Air-Korea measurements did not detect the major $NO_2$ plume as shown in Fig. 6a. Thus,
airborne remote sensing systems, such as GeoTASO, can effectively complement ground-based networks for monitoring minor
and major NOx emissions, particularly over these remote industrial regions.

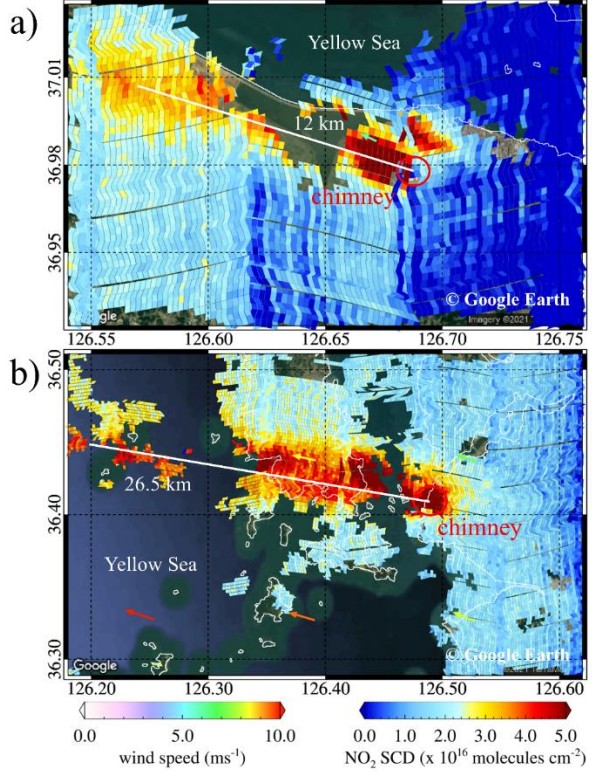


**Figure 7. Enlarged view of GeoTASO tropospheric $NO_2$ SCD observation over a) Hyundai steel works, indicated by the red box in Figure 6, and b) the Boryeong power plant, indicated by the white box in Figure 6. The arrows represent the wind direction and speed at 850 hPa from the Unified Model (UM) simulations, respectively (background RGB image is from Google Earth; https://www.google.com/maps/).**


The GeoTASO data captured not only NOx emissions from the chimneys of steelworks and power plants but also its transport
by the wind. Figure 7a and 7b show enlarged views of tropospheric $NO_2$ SCD retrieved using GeoTASO over the Hyundai
steelworks (red box in Fig. 6) and the Boryeong power plant (white box in Fig. 6). The arrows in Fig. 7 represent the prevailing
wind direction and speed from RDAPS. $NO_2$ emitted from the chimneys of these sites was transported to the Yellow Sea,
traveling distances of over 26.5 km at speeds of approximately 6 ms$^{-1}$. According to Chong et al. (2020), similar results were
found for $SO_2$ emitted and transported from these sites.
**3.1.3 Busan metropolitan region**

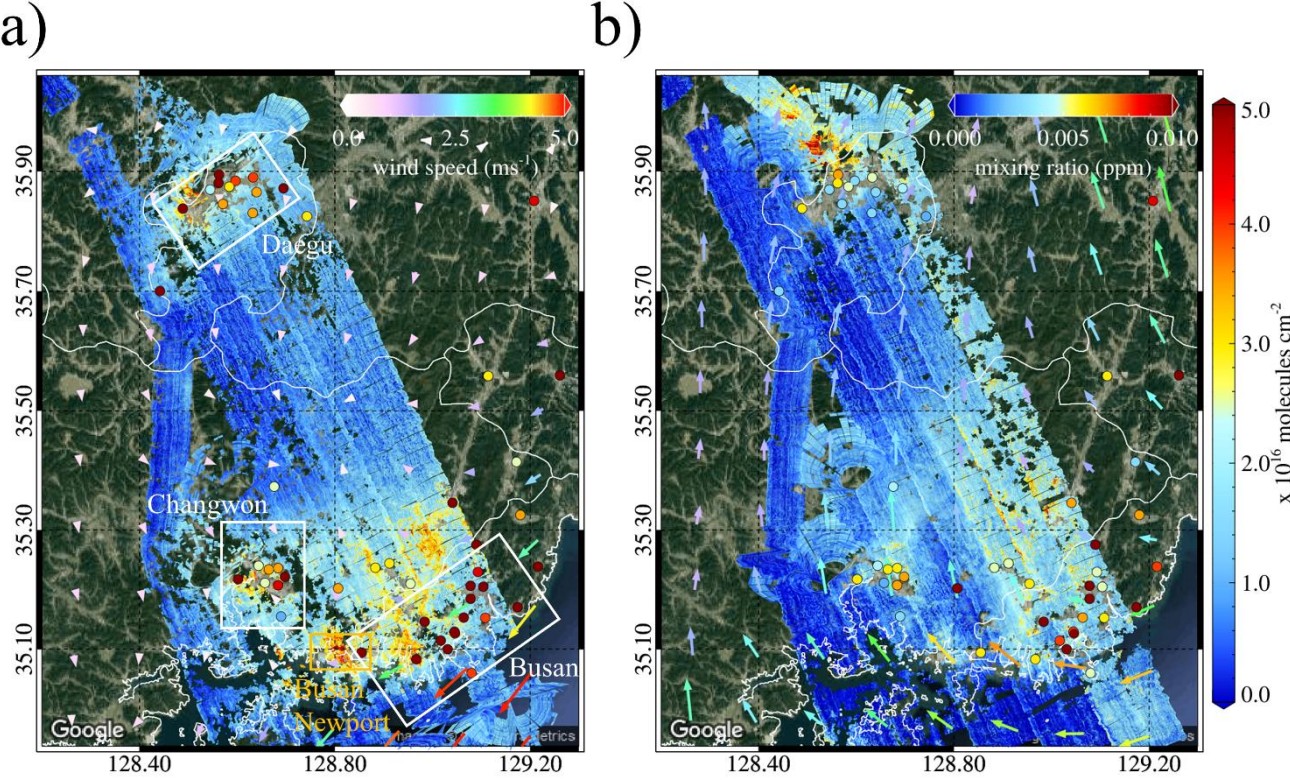


**Figure 8. Tropospheric NO₂ VCD in the Busan metropolitan region in the (a) morning and (b) afternoon of June 10, 2016.**
**The wind speed (colors scale) and wind direction (arrows) at 1000 hPa pressure level were obtained from the Unified Model (UM)**
**simulations. The white boxes represent major cities such as Busan, Daegu, and Changwon. The orange box represents Busan**
**Newport (the background RGB image is from Google Earth; https://www.google.com/maps/).**

Figure 8a and 8b show tropospheric NO₂ VCD retrieved from the GeoTASO L1B data over the Busan metropolitan region on
June 10 2016 in the morning (between 08:00 and 11:00 LT) and afternoon (between 13:00 and 16:00 LT), respectively. The
arrows in Fig. 8 indicate the wind speed and wind direction of 1000 hPa obtained from the UM-RDAPS, with the average
wind speed and wind direction of 0.9 ms⁻¹, and 55.4°, 1.9 ms⁻¹ and 147.0°, respectively, in the morning and afternoon. High
NO₂ VCDs were observed above urban areas, port, industrial complexes, and the inter-city road between Busan and Changwon.
Like the Seoul metropolitan regions, combustion using gasoline and diesel engines is estimated to contribute to the high NOx
emission. In the morning, NO₂ VCDs were high (approximately $3 \times 10^{16}$ molecules cm⁻²) in the major cities and, especially,
around Busan Newport, with values exceeding $7 \times 10^{16}$ molecules cm⁻². In comparison, in the mountainous regions between
Daegu and Busan, the NO₂ VCD values were less than $1 \times 10^{16}$ molecules cm⁻² during the same period. The spatial distribution
of tropospheric NO₂ VCDs was like that in the Seoul metropolitan regions, with high values over major cities and roads
(compare Figs. 5 and 8). In Busan, fossil fuel combustion that uses both road vehicles and ships is likely to contribute to the
NOx emissions. In the afternoon, unlike the Seoul metropolitan region, tropospheric NO₂ VCD over Busan decreased by over
$3 \times 10^{16}$ molecules cm⁻², which also corresponds with NO₂ MR data obtained from the Air-Korea sites. Detailed information
on these cities is listed in Table 3.

## 3.2 Error estimation

The accuracy of the NO$_2$ VCD retrieval using the DOAS method depends on both the AMF calculation and the spectral fitting error of the SCD retrieval. Retrieval errors of the NO$_2$ VCD were estimated using error propagation analysis as expressed in Eq. (12).

$$\frac{\varepsilon_{VCD}}{VCD} = \sqrt{(\frac{\varepsilon_{SCD}}{SCD})^2 + (\frac{\varepsilon_{AMF}}{AMF})^2}, \tag{12}$$

Where $\varepsilon_{VCD}$ is the total error of NO$_2$ VCD. The error of NO$_2$ SCD ($\varepsilon_{SCD}$) is obtained from the spectral fitting error of NO$_2$ SCD via the DOAS spectral fitting. $\varepsilon_{AMF}$ indicates the error of NO$_2$ AMF caused by uncertainties in the model input parameters for AMF calculation. Uncertainties in aerosol properties (AOD, SSA, and APH) and surface reflectance for the RTM calculations are the major factors affecting NO$_2$ AMF accuracy (Boersma et al., 2004; Leitão et al., 2010; Hong et al., 2017). Therefore, in this present study, we quantified the NO$_2$ AMF errors ($\varepsilon_{AMF}$) due to uncertainties in the input parameters independent of each other using Eq. (13):

$$\varepsilon_{AMF} = \sqrt{(\frac{\partial AMF}{\partial AOD})^2 \sigma AOD^2 + (\frac{\partial AMF}{\partial SSA})^2 \sigma SSA^2 + (\frac{\partial AMF}{\partial ALH})^2 \sigma ALH^2 + (\frac{\partial AMF}{\partial SFR})^2 \sigma SFR^2} = \sqrt{\sum_{i=1}^{4}(\frac{\partial AMF}{\partial \chi_i})^2 \sigma_{\chi_i}^2}, \tag{13}$$

where $\frac{\partial AMF}{\partial \chi_i}$ are partial derivatives of NO$_2$ AMF regarding the input parameters ($\chi_i$), $\sigma_{\chi_i}$ represents the uncertainty of the $\chi_i$. The $\sigma$ of AOD, SSA, surface reflectance, and APH are assumed to be 30% (Ahn et al., 2014), 0.04 (Jethva et al., 2014), 0.005 + 0.05 × surface reflectance (EOS Land Validation; https://landval.gsfc.nasa.gov), and 1 km (Fishman et al., 2012), respectively, in this study. To derive $(\frac{\partial AMF}{\partial \chi_i})^2$, the true $\chi_i$ is input to the RTM to simulate 'true' NO$_2$ AMF. For the AOD, SSA, APH, and surface reflectance (SFR), perturbed NO$_2$ AMF was simulated using RTM with $\chi_i \pm \sigma \chi_i$. $\partial \chi_i$ denotes the difference between the 'centre' $\chi_i$ and $\chi_i \pm \sigma \chi_i$, and $\partial AMF$ is the difference between the 'centre' NO$_2$ AMF (AMF$_{centre}$) simulated with 'centre' input values and the perturbed NO$_2$ AMF (AMF$_{perturbed}$) simulated using the perturbed input parameters $\chi_i \pm \sigma \chi_i$ (i.e. the original input parameters modified by the uncertainty). The simulation for calculating the $\varepsilon_{AMF}$ was conducted using the input parameters on 9 June 2016.

**Table 6. Total NO$_2$ VCD caused by uncertainties in NO$_2$ SCD and NO$_2$ AMF (the average for the flight on June 9, 2016).**

| | | |
|---|---|---|
| **NO$_2$ AMF errors** | AOD | 2.8% |
| | SSA | 4.1% |
| | Aerosol peak height | 22.3% |
| | Surface reflectance | 2.8% |
| | **Total NO$_2$ AMF error due to aerosol uncertainties** | **23.3%** |
| | **NO$_2$ SCD error** | **11.7%** |
| | **NO$_2$ VCD error** | **26.9%** |

Table 6 lists the estimated NO$_2$ VCD error on June 9 2016 for each source based on the error propagation method. The error estimation was conducted for the pixels where root mean square residual < 0.001 and NO$_2$ VCD > 5 × 10$^{15}$ molecules cm$^{-2}$ since NO$_2$ SCD precision is reported to be highly decreased in low NO$_2$ conditions (Hong et al., 2017). The total NO$_2$ VCD error was 26.9% with a high portion of NO$_2$ AMF error. The NO$_2$ SCD error was calculated to be 11.7%, showing the

importance of accurate DOAS spectral fitting for deriving $NO_2$ SCD. The total AMF error due to uncertainties in the input parameters was calculated to be 23.3%. Among model input parameters, the effect of APH on $NO_2$ AMF becomes high (22.3%), indicating the importance of accurate aerosol profile information. APH sensitively affects $NO_2$ AMF because near the surface where trace gases and aerosols are well mixed, aerosols lead to multiple scattering effects and the light absorption of trace gases is due to increasing light path (Castellanos et al., 2015; Hong et al., 2017). Especially, APH can be the most important input parameter in the Asia region where high loadings of aerosol plumes persist throughout the year. The $NO_2$ AMF calculation errors due to uncertainties in SSA and AOD were 4.1% and 2.8%, respectively. The $NO_2$ AMF calculation error due to uncertainties in aerosol optical properties (SSA and AOD) appears to be smaller than those in a previous study (Leitão et al., 2010). The smaller effect of the aerosol properties can be explained by the moderate aerosol loading (AOD = 0.40) on the day of flight day. The $NO_2$ AMF errors become larger under high AOD conditions. The smallest effect of SRF was found on $NO_2$ AMF calculation error, which was calculated based on the uncertainty of the SRF of the satellite-based product (MODIS). Therefore, it may be an unrealistic number for the airborne $NO_2$ AMF calculation. Once the uncertainty of airborne-based SRF is provided, considering its measurement geometry and finer spatial resolution, more realistic airborne-based $NO_2$ AMF calculation error due to uncertainties in SRF can be estimated. The can of the *a priori* $NO_2$ profile shape also be a factor to cause calculation error for $NO_2$ AMF, as reported in previous studies (Leitão et al., 2010, Meier et al., 2016, Hong et al., 2017). Therefore, it is necessary to calculate the contribution of the shape of the $NO_2$ profile *a priori* on the accuracy of $NO_2$ AMF in the future. Moreover, the resulting uncertainties of input parameters of a GeoTASO ground pixel need to be considered by combining the initial uncertainties of CTM and satellite-based products, and by the variability of the parameters within the respective CTM (AOD, SSA, and APH) and satellite (SFR) grid box. If values such as SFR are assumed constant over larger areas, the fundamental spatial variability in this these data increases the uncertainty of the AMF and hence of the determined $NO_2$ VCD on the respective finer spatial scale. In addition, the uncertainty from the assumption on the $SC_0$ and the uncertainty from ignoring the $NO_2$ above the aircraft in the AMF calculations are needed to be considered in the error analysis. This analysis should be considered in further study.

$$AMF_{percent\_change} = \frac{AMF_{perturbed} - AMF_{centre}}{AMF_{centre}} \times 100, \tag{14}$$

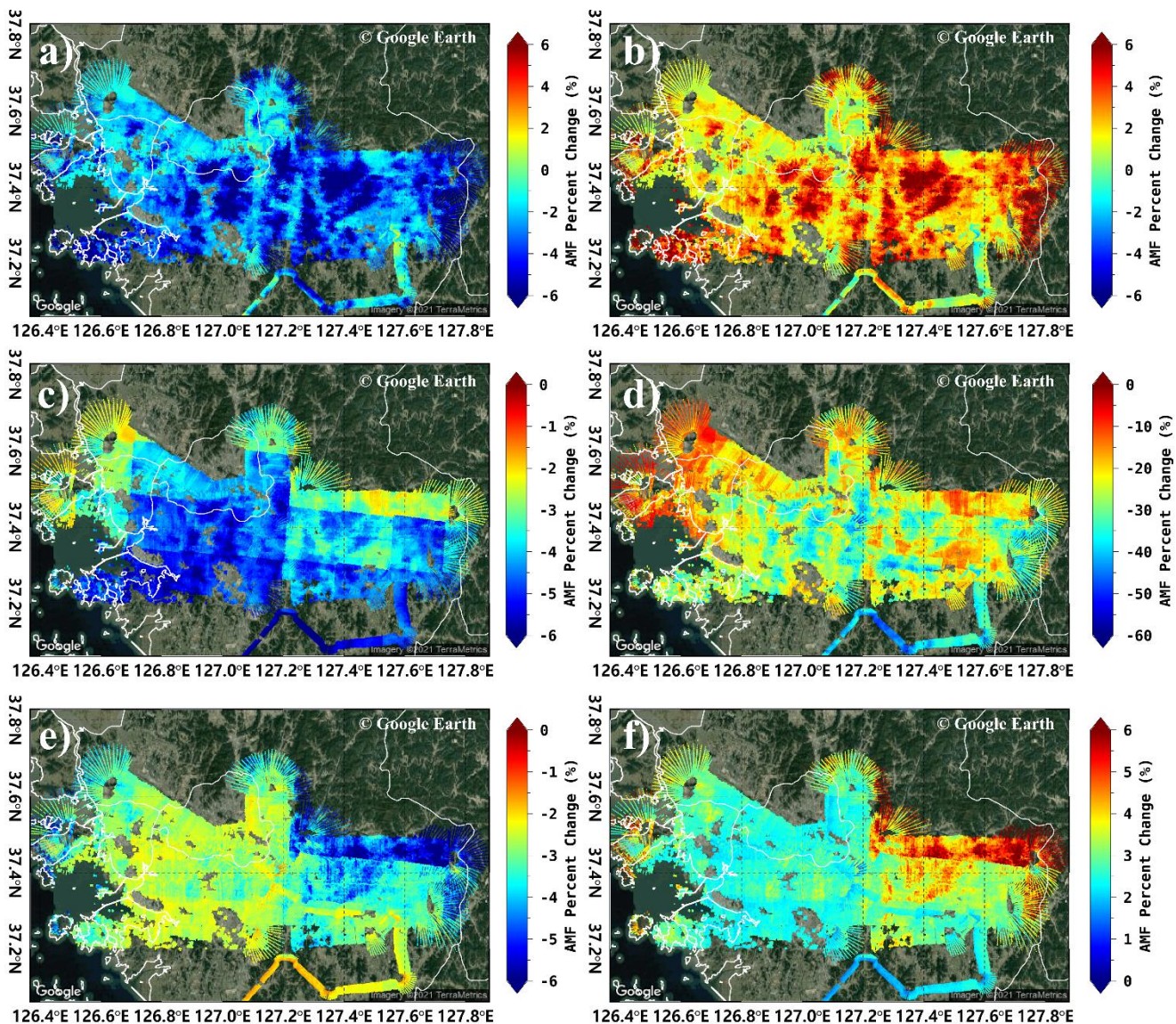

447

**Figure 9. Percent change between AMF calculated using the CMAQ model simulation and those using a) 30% lower AOD, b) 30% higher AOD, c) 0.04 lower SSA, d) 1km higher APH, compared to the model outputs. The percentage change for AMF calculated using MODIS data and those using e) 0.005 + 0.05 × SFR lower SFR, f) 0.005 + 0.05 × SFR higher SFR (background RGB image is from Google Earth; https://www.google.com/maps/).**

452

In this study, we also investigated the spatial distribution of AMF calculation errors associated with uncertainties in aerosol properties (AOD, SSA, and APH) and SFR. The percent change in $NO_2$ AMF ($AMF_{percent\_change}$) was calculated on each spatial pixel using Eq. (14). Figure 9a and 9b indicate the percentage change error between the calculated AMFs using the CMAQ AOD data with 30% lower (Fig. 9a) and 30% higher (Fig. 9b) values, respectively. The AMF decreased and increased by up to 10% with decreasing and increasing AOD, respectively, in the Seoul metropolitan region. We estimated that, under low aerosol loading conditions, an increase in AOD near the surface leads to an increase in the scattering probability within the surface layer with high $NO_2$ concentrations. Figure 9c indicates the percent change error between the calculated AMFs using CMAQ SSA data with a 0.04 lower value. The AMF decreased with decreasing SSA because the absorption of light increased. APH was also found to highly affect the accuracy of the AMF calculations (Fig. 9d). The APH uncertainty of 1 km decreased the AMFs with an average $AMF_{percent\_change}$ of -25% on the flight day. Especially, on the pixels where AOD > 0.6, the average $AMF_{percent\_change}$ was found to be -26% whereas that was -27% on the pixels where AOD < 0.4, showing the combined effect of aerosol loading and aerosol profile shape on the $NO_2$ AMF calculations. Figure 9e and 9f indicate the percentage change

error between the calculated AMFs using the MODIS surface reflectance data with $0.005 + 0.05 \times$ SFR lower (Fig. 9e) and
$0.005 + 0.05 \times$ SFR higher (Fig. 9f) values, respectively. The AMF decreased by approximately 3% when the SFR decreases,
and vice versa when it increased.

## 3.3 Validation of NO$_2$ VCDs retrieved from GeoTASO

The tropospheric NO$_2$ VCDs retrieved from GeoTASO L1B data (NO$_{2,G}$) were compared with those obtained from OMI total
NO$_2$ VCDs (NO$_{2,O}$) and Pandora (NO$_{2,P}$). The NO$_{2,O}$ were only available for June 10 during the campaign period. Therefore,
we compared only 48 NO$_{2,G}$ and NO$_{2,O}$ data points within a radius of 20 km and 30 min, which yielded a correlation coefficient
of 0.48 with a slope of 0.13 (Fig. 10 a). To validate, All NO$_{2,G}$ within a radius 20 km of the OMI center coordinate were
averaged.
The NO$_2$ values are relatively low, as GeoTASO observation is conducted in a region with low NO$_2$ compared to the Seoul
metropolitan and the overpass time of OMI is approximately 13:30 LT when NO$_2$ decreased. The low slope value is because
the OMI with low spatial resolution does not reflect the spatial NO$_2$ inhomogeneity in the pixel.

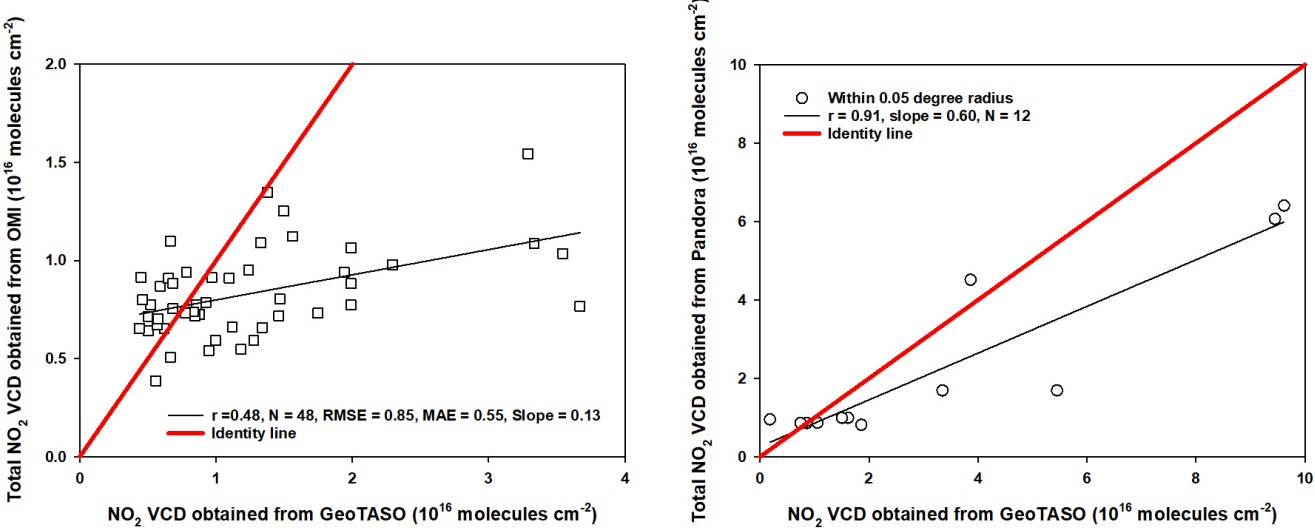


**Figure 10. Scatter plots of a) NO$_2$ VCD retrieved from GeoTASO and total NO$_2$ VCD obtained from OMI and b) total NO$_2$ VCD obtained from Pandora and NO$_2$ VCD retrieved from GeoTASO, respectively.**


To compare NO$_{2,G}$ data, we made a comparison with total NO$_2$ VCD obtained from the Pandora system (NO$_{2,P}$) during the
KORUS-AQ campaign period. NO$_{2,P}$ obtained from Busan University, Olympic Park, Songchon, Yeoju, and Yonsei University
Pandora sites on June 5, 9, and 10 were used for the GeoTASO validation (Fig. 1). NO$_{2,G}$ and NO$_{2,P}$ columns at these sites are
compared in Fig. 11. To compare NO$_{2,G}$ and NO$_{2,P}$, we used averaged NO$_{2,G}$ retrieved from 16 across tracks with the smallest
viewing zenith angle and averaged 30 min NO$_2$ obtained from pandora measurement within a radius of approximately 0.05°.
NO$_{2,G}$ and NO$_{2,P}$ were correlated (R = 0. 91, with a slope of 0.60), however, when NO$_{2,P}$ was lower than $1 \times 10^{16}$ molecules
cm$^{-2}$, the correlation coefficient between NO$_{2,G}$ and NO$_{2,P}$ was < 0.1. The weak correlation at low NO$_2$ levels most likely
reflects differences in viewing geometries and the horizontal inhomogeneity of the measured NO$_2$ between Pandora and
GeoTASO. Furthermore, Pandora and GeoTASO can be used for the NO$_2$ validation of geostationary satellites, such as GEMS.
However, because the number of pandora is limited in this campaign, we difficulty validating NO$_2$ retrieved from GeoTASO
under various conditions. Many ground-based remote sensing measurements are needed to validate GEMS under various
conditions.



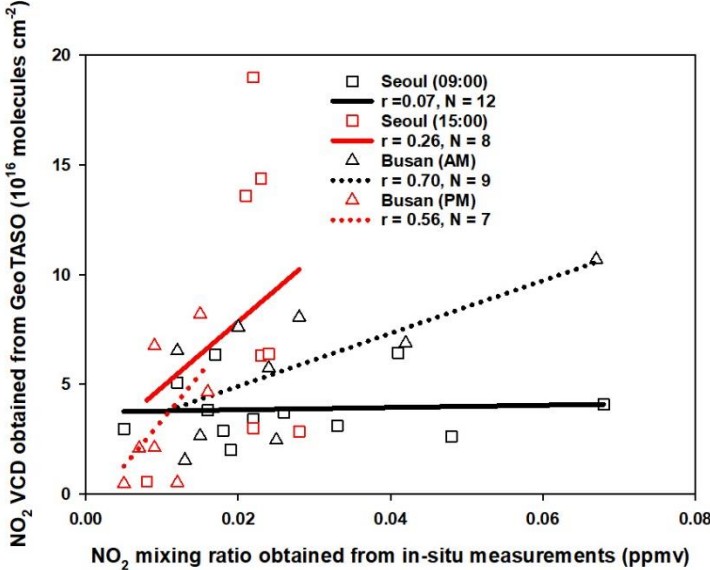

**Figure 11. Scatter plot of the NO₂ VCDs retrieved from GeoTASO, and NO₂ surface mixing ratio obtained from Air-Korea. The black and red squares represent the NO₂ data at 9 AM and 3 PM (local time) in the Seoul metropolitan region, respectively. The black and red triangles represent those in the morning and afternoon, over Busan, respectively.**

To compare the spatiotemporal distribution of NO₂ VCDs retrieved from GeoTASO, $NO_{2,G}$ compared with surface spatial patterns, $NO_{2,G}$ was compared with $NO_{2,A}$ for GeoTASO data within a radius of approximately 0.05 km and 30 min (Fig. 11). To compare $NO_{2,G}$ and $NO_{2,A}$, we used averaged $NO_{2,G}$ retrieved from 16 across tracks and averaged 30 min within a radius of 0.05°. Because in situ measurements provide $NO_2$ VMR ($NO_{2,A}$)(ppmv) once per hour, $NO_{2,A}$ of the nearest time is used to compare with $NO_{2,G}$. The correlation coefficient (R) between $NO_{2,G}$ (molecules cm$^{-2}$) and $NO_{2,A}$ at 9 AM and 3 PM LT in the Seoul metropolitan region was 0.07 and 0.26, respectively. When using only roadside station data from Air-Korea, the R-value for the morning increased to 0.72, which implies GeoTASO is more sensitive to emissions from $NO_2$ source areas, such as roadsides (Fig. 5). Because the comparison, there were large differences in the morning and afternoon. These results were identified because synoptic meteorology played an important role from June 1 to June 10, 2016 (Choi et al., 2019). As described by Judd et al. (2018), the spatial distribution for $NO_2$ VCDs appears to reflect the emission source in local industrialized regions and transportations in the morning with relatively weak winds. $NO_2$ concentration often increases in the late morning, indicating that the emission process proceeds faster than the $NO_2$ removal process. As the planetary boundary layer heights (PBLH) in early afternoon increase and surface $NO_2$ is mixed through a deeper PBLH, the $NO_2$ VCDs distribution showed a wider increase in most of the Seoul metropolitan area and the column amounts continue to increase (Judd et al., 2018).

When comparing $NO_2$ VCDs with surface $NO_2$ concentrations, it should be highlighted that it is a nonlinear relationship between $NO_{2,G}$ and $NO_{2,A}$. Although it may vary depending on weather conditions, high $NO_2$ VCDs from airborne observations can sometimes be detected with low surface $NO_2$ concentrations. When exhaust gases emitted from industrial facilities occur at a certain altitude (stacks/chimneys), $NO_{2,G}$ show high $NO_2$ VCDs, but $NO_{2,A}$ may be observed to have a low concentration. Unfortunately, in the Anmyeon industrial region, $NO_{2,G}$ and $NO_{2,A}$ could not be compared due to spatial restrictions because the distribution of ground observation stations is concentrated in metropolitan areas.

In the Busan metropolitan area, the R-value of the $NO_{2,G}$ and $NO_{2,A}$ data had a correlation coefficient greater than 0.56. This reflects the more even horizontal distribution of $NO_2$ in the afternoon, when diffusion from the source areas had occurred. However, for a more accurate comparison, $NO_2$ VCD data should be converted to $NO_2$ MR based on mixing layer height, temperature, and pressure profile data (Kim et al., 2017; Qin et al., 2017; Jeong and Hong, 2021a). However, because the number of pandora and satellite data is limited in this campaign, we had difficulties in validating $NO_2$ retrieved from GeoTASO

under various conditions. Because ground-based, airborne and space borne remote sensing measurements have their own
advantages and disadvantages, it is recommended a comprehensive observation campaign involving all of ground-based,
airborne and space-borne measurements should be conducted continuously for the upcoming new era of geostationary
environmental satellites.

## 4 Conclusions

For the first time, we have retrieved $NO_2$ VCD data using airborne GeoTASO observations over the Seoul metropolitan
region—one of the most populous cities worldwide, the Busan metropolitan region—the second-largest city in South Korea,
and Anmyeon, with thermal power plants and industrial complexes. By retrieving $NO_2$ data using GeoTASO L1B radiance, it
was possible to observe the spatial distribution of $NO_2$ in these metropolitan and industrial regions. In the morning,
tropospheric $NO_2$ VCD in Seoul showed a strong horizontal gradient between rural and urban areas. In urban areas,
tropospheric $NO_2$ VCD was high, with values exceeding $3 \times 10^{16}$ molecules cm$^{-2}$; in rural areas, values were typically below
$1 \times 10^{16}$ molecules cm$^{-2}$. Extremely high values over $10 \times 10^{16}$ molecules cm$^{-2}$ were also observed in both rural and urban
areas. In Anmyeon, GeoTASO observations showed that $NO_2$ is mainly emitted from the chimneys of industrial complexes
and thermal power plants, and subsequently transported by wind approximately 30 km to the Yellow Sea on the west coast of
the Korean Peninsula. In the Busan metropolitan region, in the morning, tropospheric $NO_2$ VCDs showed a pattern similar to
the Seoul metropolitan region, with high values above the inter-city road. However, unlike Seoul, tropospheric $NO_2$ VCDs in
Busan decreased in the afternoon due to local different weather conditions locally.
To compare the data retrieved from the GeoTASO system, we compared $NO_{2,G}$ with $NO_{2,O}$ obtained from the OMI, $NO_{2,A}$
obtained from Air-Korea, and $NO_{2,P}$ obtained from the Pandora observation system. When the distance between two
observations was below 20 km or 0.05° within 30 min, the correlation coefficients were relatively high (R = 0.48, and 0.91,
respectively). However, the correlation between $NO_{2,G}$ and $NO_{2,A}$ over the Seoul metropolitan region was extremely weak (R
= 0.07) in the morning because of the more pronounced $NO_2$ horizontal gradient.
The GeoTASO system successfully observed $NO_2$ VCDs with high horizontal spatial resolution for both metropolitan and
industrial regions. This demonstrates that airborne remote sensing measurements from GeoTASO, similar to GCAS, APEX
and others, can be an effective tool for the validation of trace gases retrieved from environmental satellites, including the OMI,
TROPOMI, and GOME-2; these systems can obtain high-resolution measurements over relatively wide areas. However, to
validate geostationary environmental satellites with higher spatiotemporal resolutions, such as the GEMS, TEMPO, and
sentinel-4, additional validation strategies are needed. Based on error estimation, it can be concluded that aerosol properties
are relevant and should be determined and $NO_2$ vertical profile retrieval performed using, for example, LIDAR, MAX-DOAS,
and sondes. This is important because the accuracy of aerosol properties, surface reflectance and the $NO_2$ vertical profiles
affects the accuracy of AMF calculations (Leitão et al., 2010; Hong et al., 2017; Lorente et al., 2017; Boersma et al., 2018).
Furthermore, as we observed in the Seoul metropolitan area, closer spaced observations using ground-based remote sensing
systems and in situ measurements are needed as $NO_2$ displays large horizontal gradients, especially in the morning.

## Author contributions

**GH** and **HH** designed and implemented the research. **KL** provided the CTM data. **GH** developed the code for model running
and performed the RTM simulations. **HH** and **UJ** contributed to the analysis of ground-based data. **GH** and **WC** conducted
the sensitivity test. **GH**, **KL**, **HH**, **UJ**, **WC,** and **JJS** revised and edited the paper. **HH**, **UJ, and WC** provided constructive
comments. All authors contributed to this works.
**Competing interests**
The authors declare that they have no conflict of interest.
**Acknowledgements**
Pandora data were obtained from the KORUS-AQ home pages of NASA's Goddard Space Flight Center
(https://avdc.gsfc.nasa.gov/pub/DSCOVR/Pandora/DATA/KORUS-AQ/). Ground-based $NO_2$ MR data were obtained
from Air-Korea (http://www.airkorea.or.kr/web/detailViewDown?pMENU_NO=125/). The authors would like to
thank KORUS-AQ campaign team for providing the GeoTASO and Pandora data.
**Funding**
This work was funded by the National Institute of Environmental Research (NIER) of the Ministry of Environment [No. NIER-
2021-01-01-100].

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
