# Peer review of "Highly resolved mapping of NO2 vertical column densities from"

_Atmospheric Measurement Techniques, 2022_

## Referee Comment (RC1)

**Review of: Highly resolved mapping of NO$_2$ vertical column densities from GeoTASO measurements over a megacity and industrial area during the KORUS-AQ campaign** (Choo et al., 2022)

The manuscript discusses results from the KORUS-AQ campaign and demonstrates for the first time the NO$_2$ VCD horizontal distribution over highly urbanized / industrialised regions in South Korea, based on airborne imaging data. The scientific content of the paper fits within the scope of AMT, although the study lacks novelty. The paper is short on the actual data retrievals and focuses more on the interpretation of the NO$_2$ spatiotemporal variability over the different regions. Regarding the VCD retrievals, a number of choices/assumptions are made without proper argumentation which are not consistent with other similar studies that included extensive explanations/analysis. This can affect the results and the made conclusions, especially regarding the absolute VCD values. I'm referring for example to the fitting interval, reference SCD, surface reflectance product. Also the error analysis has some serious flaws and the 'validation' lacks a lot of details in order to assess the validity of the comparisons. Overall, I would recommend publication in AMT. However, major revisions (detailed below) need to be conducted in the paper before publication. Please note that all questions below should not only be addressed in the author's reply, but also in the manuscript.

**General comments**

-Please extend section 2.1. with campaign information. Some information about the campaigns is scattered in the manuscript, but a clear campaign section shortly discussing the number of flights, time and duration of flights, ROI, SZA change during flights, environmental conditions, e.g. cloud fraction, etc. is missing while it would improve interpretation. Also a table would help indicating the different flights, their time of flight and the region of interest. This would also help to understand certain choices made for the comparisons (see later comments on that).

-p.5 l.150: One reference is used to analyse the whole data set if I understand it correctly. Three questions that should be properly addressed: 1) do you use one spectrum or an average of spectra over your reference area in order to improve the SNR? ; 2) What is the residual amount in the reference spectrum used and how was this determined. Do you use the value 6.751 x 1015? This is not really clear from the manuscript. This value seems high for a considered background area over the ocean and can result in an overestimation of all retrievals shown in this work. Have you compared with the OMI value retrieved for that day or an average + st. dev. OMI value to get an idea on the NO$_2$ variability for this area? ; 3) for airborne hyperspectral imagers often a daily reference is preferred as the spectral properties of the instrument can change resulting in an along-track and across-track drift. Please prove that this is not the case with this instrument and the right choice was made, i.e. by showing RMS of the fit on the day when the reference was taken and RMS for flights further away from the reference date.

-p.5 l.155: This NO$_2$ spectral window is not very optimal and moreover very narrow, while the instrument properties allow a more optimal and larger window for NO$_2$ fits, e.g. like the 425-490 nm recommended in the literature. Please show some fit results with this larger window (or similar larger window) and show the VCD differences with the narrow window to clarify the impact of the choice made and also give a clear motivation for choosing this non-optimal window 425-450 nm for your retrievals.

p.6 l.175: It is shown in several studies, some of them cited in this work, that surface reflectance can have a very strong impact on the AMF. Is GeoTASO not absolutely calibrated in order to derive a surface

reflectance product from the instrument itself? If not, why isn't the MODIS product MCD43A1 or MCD43A3 used like during the LISTOS campaign? It provides a much higher spatial resolution matching better the spatial resolution of your retrievals and it is proven to be a reliable product. I'm worried that the coarse resolution product you have chosen strongly impacts your AMF, thus also your VCD retrievals.

Sect. 2.5: You clarified later on that SSA and AOD are also derived from CMAQ. This should be described here as well, besides the vertical $NO_2$ profile.

Sect 3.1: The spatial binning (0.01°) has not been explained in the manuscript, but only mentioned in the relevant figure captions. Do you bin spectra prior to the retrievals to increase the SNR or do you bin the VCDs afterwards? In the latter case, why do you bin the data (native resolution of 250 m) to this coarser resolution? You are losing some spatial detail while $NO_2$ is a species with strong spatiotemporal variability.

p.7 l.204 and Figure. 4: I would like to see a zoom on figure 4 on the expressways (eventually at the native spatial resolution). In Figure 4, it is difficult to judge if there is a stronger $NO_2$ pattern downwind than upwind from the expressway line sources, but this is something that should be present in the data if the made statements are correct. You could also do it for the afternoon with a different dynamic scale for the VCD map.

P.7 l.206: So it means the background $NO_2$ VCD is around 1 x $10^{16}$? This seems pretty high (see also earlier comment on the SCDref). Can you demonstrate that this is somehow consistent with OMI/TROPOMI retrievals? Please do the same for statement on p. 9 l. 255

p.9 l.364: The error on the SCDref is an important source of uncertainty as well, not taken into account in this study. Please include.

p.9 l.273: Please clarify how these percentages were obtained. This is unclear. An important uncertainty regarding the AMF is also the uncertainty related to the a priori $NO_2$ profile shape. This is mentioned as well in the conclusion but it is completely ignored in this uncertainty study. Please include.

p.10 l.283: 7.3% uncertainty on the AMF is really on the optimistic side and not in line with other studies that report around 15%-20%, or larger. Please revise your calculations and/or explain. Just to clarify: it is perfectly fine if you obtain other numbers than in other studies but this should be clarified based on an in-depth analysis. Like mentioned in the previous comment it is unclear where the percentages are coming from, while in other studies these are based on discussed sensitivity tests. Are the numbers provided in Table 6 an average for the flight on 9 June 2016 or what does it represent exactly?

Sect. 3.3. all the performed comparisons should report how many measurements/data points are compared (eventually in the plots), in order to assess how statistically relevant the comparisons are.

p.10 l.308: Elaborate on this part, eventually with a plot, as many things are unclear! Why are OMI NO2 VCDs only available on 10 June? How did you perform the comparison exactly? Setting a radius is a valid strategy when comparing ground based measurements with airborne or spaceborne data, but not when comparing airborne with satellite. In the latter case you should average the airborne pixels within the larger satellite pixel footprints in order to perform a fair comparison. The time constraint can be kept but could be extended to 1 hour to increase amount of compared measurements. Slope is 0.43: not clear without plot what is under- / overestimating what.

Sect. 3.3.1: first sentence should be rewritten as it is unclear. Again not clear: do you compare all pixels within 0.5 km and 30 min with the station measurement or do you average the airborne pixels and do you compare the averaged value? You have hourly station measurements and many flights over the whole campaign period (I think? Much details are missing on flights), but you don't seem to have many data points. Please clarify this. Why not focusing on all possible comparisons between airborne and ground-based measurements instead of restricting to AM and PM in order to improve your statistics. Like it is presented now: not really statistically relevant and without any proper discussion on the non-linear relation between columns and surface concentration, I suggest to leave this section out of the paper.

p.11 l.325: Why do you restrict to 5, 9, 10 June? You should clarify the reasons. If you only had flights over this region on these dates that is a proper explanation, but these details have not been provided.

Figure 5: I have a hard time to understand the AMF you obtained. Normally the AMF should be highly correlated with the surface reflectance which isn't really the case here. One (partly) explanation is maybe the rough albedo product used. But this doesn't explain the strong striping in the AMF, especially in the western part. Can you elaborate on this in the manuscript to explain the reasons?

Figure 10: I would expect that PANDORA retrievals are higher than the GeoTASO retrievals as the ground-based measurements are more sensitive to the bulk of NO2 close to the surface, but it is the other way around. This could be related to the choices made in the VCD retrievals like mentioned in earlier comments which can strongly affect the absolute values. Are these findings consistent with other GeoTASO-PANDORA comparisons like from the DISCOVER-AQ or LISTOS campaign?

**Minor comments**

p.3 l.79: SWING has not been operated over Antwerp. The example of Antwerp shown is a simulation. SWING has been operated on a UAV over Romania. It is also better to refer to https://amt.copernicus.org/articles/11/551/2018/amt-11-551-2018.pdf

p.3 l.81: "regional radiative transfer models" → I think you are referring to regional air quality models here?

p.3 l.89: How is transboundary pollution defined here?

p.4 l.120: Be more clear on how exactly you do the comparison: if an airborne overpass matches the 30 min constraint, do you average spatially all the pixels within 1 km radius? Or do you compare the different individual pixels with the Pandora retrieval? Same comment for p. 5 l. 131.

p.5 l.143: What is the exact reason of flying so high to measure tropospheric species? At 3 km you would already be well above the PBL. And while indeed the higher flight altitude allows to measure the NO2 in the free troposphere as well, it has the drawback that you are losing some sensitivity towards the NO2 in the PBL and especially at the surface due to larger scattering and absorption probability.

p.7 l.210: Also the PBL plays a role and it's rise in the afternoon + transport/accumulation of emitted $NO_2$ in the PBL during the day. But the latter is already stated later on in this paragraph.

p.8 l.240: I agree with the statement that they can be highly complementary, but it should be clarified to the reader that the relation VCD-surface concentration is highly non-linear. Depending on meteorology it is possible that strong columns are detected while surface concentrations are low. Especially in case of industrial emissions, where the emissions happen at a certain altitude (stacks/chimneys).

p.13 l.376: Link doesn't work

Table 2 and 5: Average mileage is average mileage per car, per day?

**Technical corrections**

p.1 l.24: please replace 'data' by 'VCDs'

p.2 l.31: you might add 'domestic heating' as well

p.2 l.38: cites --> cities

p.2 l.45: what is the point of mentioning UTC here if you don't specify a time?

p.2 l.53: is there a need to repeat here the spaceborne sensors? This was done in the previous paragraph

p.3 l.90: replace "from May to June 2016." By "…, organized from May to June 2016."

p.3 l.94: In this study, NO2 VCD retrieval was conducted using solar backscattered radiance observed from GeoTASO over South Korea during the KORUS-AQ campaign --> This sentence is redundant, please remove.

p.4 l.106: Amnyeon --> Amnyeon region?

p.4 l.118: O3 → $O_3$

p.4 l.122: ". " after Notably

p.5 l.132: "." at end of sentence.

p. 6 l.184: stratospheric NO2 → stratospheric and free tropospheric $NO_2$

p. 8 l.219: decrease --> decreases

p. 9 l.268: Boersma et al. 2004 --> Boersma et al., 2004

Figure 1 caption: please repeat the campaign period here and the number of flights.

Figure 3 caption: "." at end of sentence.

Figure 4 caption: remove 'the' in 'the each panel'

Figue 6 caption "into to" --> please correct + what is the meaning of the color-coding of the arrows?

---

## Author Comment (AC1)

The manuscript discusses results from the KORUS-AQ campaign and demonstrates for the first time the NO2 VCD horizontal distribution over highly urbanized / industrialised regions in South Korea, based on airborne imaging data. The scientific content of the paper fits within the scope of AMT, although the study lacks novelty. The paper is short on the actual data retrievals and focuses more on the interpretation of the NO2 spatiotemporal variability over the different regions. Regarding the VCD retrievals, a number of choices/assumptions are made without proper argumentation which are not consistent with other similar studies that included extensive explanations/analysis. This can affect the results and the made conclusions, especially regarding the absolute VCD values. I'm referring for example to the fitting interval, reference SCD, surface reflectance product. Also the error analysis has some serious flaws and the 'validation' lacks a lot of details in order to assess the validity of the comparisons. Overall, I would recommend publication in AMT. However, major revisions (detailed below) need to be conducted in the paper before publication. Please note that all questions below should not only be addressed in the author's reply, but also in the manuscript.

**General comments**

-Please extend section 2.1. with campaign information. Some information about the campaigns is scattered in the manuscript, but a clear campaign section shortly discussing the number of flights, time and duration of flights, ROI, SZA change during flights, environmental conditions, e.g. cloud fraction, etc. is missing while it would improve interpretation. Also a table would help indicating the different flights, their time of flight and the region of interest. This would also help to understand certain choices made for the comparisons (see later comments on that).

**Response:** First of all, we sincerely apologize for the late submission.

We thank the reviewer's kind comment and advice. Additional descriptions of the campaign and flight information (number, time, altitude, route, and etc.) have been added into the manuscript. Please, see P. 5, Line 125. The detailed information on each flight date is on the http://www-air.larc.nasa.gov/missions/korus-aq/docs/KORUS-AQ_Flight_Summaries_ID122.pdf as well as written in the manuscript.

-p.5 l.150: One reference is used to analyse the whole data set if I understand it correctly. Three questions that should be properly addressed: 1) do you use one spectrum or an average of spectra over your reference area in order to improve the SNR? ; 2) What is the residual amount in the reference spectrum used and how was this determined. Do you use the value 6.751 x 1015? This is not really clear from the manuscript. This value seems high for a considered background area over the ocean and can result in an overestimation of all retrievals shown in this work. Have you compared with the OMI value retrieved for that day or an average + st. dev. OMI value to get an idea on the NO2 variability for this area? ; 3) for airborne hyperspectral imagers often a daily reference is preferred as the spectral properties of the instrument can change resulting in an along-track and across-track drift. Please prove that this is not the case with this instrument and

**the right choice was made, i.e. by showing RMS of the fit on the day when the reference was taken and RMS for flights further away from the reference date.**

Response: In this present study, we used reference observed radiance averagedby 250 m × 250 m in each 33 across tracks over South Ocean of Jeju Island wihich is one of the most clean region in south Korea. The NO2 VCD and standard deviation in OMI in this region (SZA<85, cloud fraction < 0.5) is 4.77 × $10^{15}$ molec. cm$^{-2}$ and 1.33 × $10^{15}$ molec. cm$^{-2}$, respectively. When radiance over the high NO$_2$ region is used as a reference, the fitting results normally are underestimated. I think you see the NO$_2$ VCD is overestimated in Seoul metropolitan regions especially background regions in the afternoon. But, the NO$_2$ VCD has a very high value in GEMS and sometime those has large values in the afternoon than morning time (Left, 3, May, 2021, 09 UTC: right, 3, May, 2021, 12 UTC).

[Figure]

As you mentioned, I agree that it is good to set a reference at each flight, but as you can see in figure 1, most Geo-TASO observation do not cover the clean area.

**-p.5 l.155: This NO2 spectral window is not very optimal and moreover very narrow, while the instrument properties allow a more optimal and larger window for NO2 fits, e.g. like the 425-490 nm recommended in the literature. Please show some fit results with this larger window (or similar larger window) and show the VCD differences with the narrow window to clarify the impact of the choice made and also give a clear motivation for choosing this non-optimal window 425-450 nm for your retrievals.**

Response: We also agree with your comment that NO$_2$ VCD resulted from fitting using a wide spectral window (ex, 425 ~ 490 nm) is more stable, especially space borne measurement. But in this present study, we prepared all dataset (a priori AOD, airmass factor calculation) with spectral

window from 425 nm and 460 nm. I compared the retrieved NO$_2$ VCD of 16 cross tracks in 5, June between spectral window from 425 ~ 460 nm and those from 425 ~ 490 nm as follows:

[Figure]

If you think that NO$_2$ VCD should be retrieved by using fitting window between 425 nm and 490 nm, we will perform AOD re-calculation by using CMAQ and also we will re-calculate AMF.

**p.6 l.175: It is shown in several studies, some of them cited in this work, that surface reflectance can have a very strong impact on the AMF. Is GeoTASO not absolutely calibrated in order to derive a surface reflectance product from the instrument itself? If not, why isn't the MODIS product MCD43A1 or MCD43A3 used like during the LISTOS campaign? It provides a much higher spatial resolution matching better the spatial resolution of your retrievals and it is proven to be a reliable product. I'm worried that the coarse resolution product you have chosen strongly impacts your AMF, thus also your VCD retrievals.**

**Response**: We agree with reviewer's comments, thus tried our best to answer your questions. First of all, as you mentioned, errors due to AOD, SSA, surface reflectance, etc., which are used as input data when calculating AMF, may affect AMF results. Therefore, in this study, the uncertainty according to each input variable was calculated and provided in Section 3.2. In our result, the variable with the largest AMF error was aerosol loading height (26.4%), which is SSA (4.2%), AOD (3.0%), and surface reflectance (2.8%). In addition, the average of the surface reflectance during

KORUS-AQ period is 0.055, and there is a study using a fixed ground reflectance of 0.05 to retrieve SO$_2$ VCD (Chong et al., 2020).

GeoTASO does not make any corrections to calculate surface reflectance from the device itself. However, GeoTASO data for NO$_2$ retrieval were spectrally, geometrically, radiometrically calibrated at the NASA Goddard Space Flight Center. The algorithms for calculating observed surface reflectance using GeoTASO data and surface reflection results were not provided. We tried to use surface reflectance (500 m resolution) of MCD43A3, but there were many pixels that were not produced within the observation area. We wanted to know the spatial distribution of NO$_2$ VCD. Therefore, we used MOD09CMG and MYD09CMG. Because the data exist in all pixels, the surface reflectance.

♦   Chong, H., Lee, S., Kim, J., Jeong, U., Li, C., Krotkov, N. A., Nowlan, C. R., ... & Koo, J.-H. (2020). High-resolution mapping of SO2 using airborne observations from the GeoTASO instrument during the KORUS-AQ field study: PCA-based vertical column retrievals. Remote Sensing of Environment, 241, 111725.

**Sect. 2.5: You clarified later on that SSA and AOD are also derived from CMAQ. This should be described here as well, besides the vertical NO2 profile.**

**Response**: As reviewer suggested, sentences and equations explaining the values (SSA, AOD, and vertical NO$_2$ profiles) have been added to the revised manuscript in P. 10, Lines 237-246.

**After modification (P. 10, Lines 237-246):**

"CMAQ AOD was calculated by integrating the aerosol extinction coefficient ($Q_{ext}$), which is the sum of scattering ($Q_{sca}$) and absorption ($Q_{abs}$) coefficients, over all vertical layers ($z$) as follows:

$$AOD = \int Q_{sca}(z)\, dz = \int \{Q_{sca}(z) + Q_{abs}(z)\}\, dz \tag{9}$$

$$Q_{abs}[\text{Mm}^{-1}] = \sum_i \sum_j \{(1 - \omega_{ij}) \cdot \beta_{ij} \cdot f_{ij}(RH) \cdot [C]_{ij}\} \tag{10}$$

$$Q_{sca}[\text{Mm}^{-1}] = \sum_i \sum_j \{\omega_{ij} \cdot \beta_{ij} \cdot f_{ij}(RH) \cdot [C]_{ij}\} \tag{11}$$

Here, $\omega_{ij}$ indicates SSA of particulate species i for the particulate mode (or size bin) j, $\beta_{ij}$ denotes the mass extinction efficiency, $f_{ij}(RH)$ is the hygroscopicity factor at the relative humidity (RH), and $[C]_{ij}$ is the concentration of particulate species. CMAQ SSA is defined as the ratio of the integrated $Q_{sca}$ to AOD, and NO2 vertical profiles were obtained from NO2 concentrations at each vertical layers by conducting CMAQ simulations. More details of the model descriptions are shown in Lee et al. (2020) and Malm and Hand (2007)."

♦   Malm, W. C. and Hand J. L.: An examination of the physical and optical properties of aerosols collected in the IMPROVE program, Atmospheric Environment, 41, 3407– 3427, https://doi.org/10.1016/j.atmosenv.2006.12.012, 2007.

**Sect 3.1: The spatial binning (0.01°) has not been explained in the manuscript, but only mentioned in the relevant figure captions. Do you bin spectra prior to the retrievals to increase the SNR or do you bin the VCDs afterwards? In the latter case, why do you bin the data (native resolution of 250 m) to this coarser resolution? You are losing some spatial detail while NO2 is a species with strong spatiotemporal variability.**

**Response:** We apologize for the confusion. We agree with the reviewer's advice and have therefore revised to the manuscript in P. 10, Line 249. We did spatial binning after calculating $NO_2$ VCD. In Chong et al. (2020), we have $SO_2$ VCD by using data observed in GeoTASO during the KORUS-AQ period, and binning was performed at a resolution of GEMS (7 x 8 km$^2$). As a result of previous studies, it was found that relative random uncertainty decreased as $SO_2$ VCD increased. This random uncertainty can be reduced even when additional spatial binning is performed, which provides a balance between random error and spatial resolution. In conclusion, the study showed that larger VCDs at 250 m resolutions do not necessarily lead to larger VCDs at 7 km x 8 km resolutions. Additionally, we showed the AMF which is native resolution by modifying Fig. 5(c). Responding to your comments.

**After modification (P. 10, Line 249-252):**

"We showed the finally $NO_2$ VCDs by binning them with $0.01° \times 0.01°$ from 250 m spatial resolution. Although the spatial binning $NO_2$ VCDs were compared to those at native resolution, we noted that the spatiotemporal variability was still able to be clearly distinguished from the background at $0.01°$ binning resolution. Chong et al. (2020) showed that larger VCDs at 250 m resolutions do not necessarily lead to larger VCDs at wider resolutions."

- Chong, H., Lee, S., Kim, J., Jeong, U., Li, C., Krotkov, N. A., Nowlan, C. R., ... & Koo, J.-H. (2020). High-resolution mapping of SO2 using airborne observations from the GeoTASO instrument during the KORUS-AQ field study: PCA-based vertical column retrievals. Remote Sensing of Environment, 241, 111725.

**p.7 l.204 and Figure. 4: I would like to see a zoom on figure 4 on the expressways (eventually at the native spatial resolution). In Figure 4, it is difficult to judge if there is a stronger NO2 pattern downwind than upwind from the expressway line sources, but this is something that should be present in the data if the made statements are correct. You could also do it for the afternoon with a different dynamic scale for the VCD map.**

**Response:** We have indicated the $NO_2$ VCD with the native resolution (250 m) in Fig. R1 responding to your comments. At this time, the VCD map (Fig. 4(b)) in the afternoon showed $NO_2$ VCD with a different dynamic scale than before. Additionally, (2) Seohaean expressway in Fig. 4(a) is enlarged and represented at the same color scale of the same $NO_2$ VCD (Fig. R2). In the enlarged Fig. R2, it was found that the NO2 VCD was relatively higher on the highway than on the surrounding mountains. We showed wind direction and wind speed at 1000 hPa level in the morning and afternoon respectively. It can be seen that the wind directions are different in the morning and afternoon on 9, June 2016, and the wind speed is relatively higher in the afternoon than in the morning.

[Figure]

**Fig. R1.** Similar to Fig. 4 except that the native spatial resolution (250 m).

[Figure]

**Fig. R2.** A zoom in Fig. R1-(2) Seohaean expressway (the red line).

**P.7 l.206: So it means the background NO2 VCD is around 1 x 1016? This seems pretty high (see also earlier comment on the SCDref). Can you demonstrate that this is somehow consistent with OMI/TROPOMI retrievals? Please do the same for statement on p. 9 l. 255**

**Response:** I understand your opinion that the background $NO_2$ VCDs are very high. And unfortunately, in this period, TROPOMI was not launched yet, OMI did not provide $NO_2$ data except for a very few days. We re-retrieved $NO_2$ VCDs during review period, but $NO_2$ VCDs still remained high in Seoul metropolitan region. But as you can see Goldberg et al., 2019 this figure 6, the averaged $NO_2$ VCDs in May, 2016 retrieved OMI have large values in Seoul metropolitan region, and also they have larger values over 3 x $10^{16}$ molec. $cm^{-2}$ when AMF is adjusted. It is irrelevant with this paper, but sometime GEMS NO2 values are extremely large in Seoul metropolitan region, especially in the afternoon. We also validate this large values by using ground-based measurements in GEMS Map of

Air Pollution (GMAP) and Satellite Integrated Joint Monitoring Air Quality (SIJAQ) campaign which has been carried out since 2019. I also think it is necessary to keep finding the reason why $NO_2$ VCDs increase in the afternoon in Seoul metropolitan region although $NO_2$ VMR observed from in-situ measurements decreases.

[Figure]

**Figure 6. (a)** Total vertical column contents from the OMI-standard $NO_2$ product for May 2016, **(b)** same quantities from the OMI-regional product with only the air mass factor adjustment (AMF) during the same time frame, **(c)** same quantities from the OMI-regional product with the air mass factor adjustment and spatial kernel (AMF + SK) during the same time frame, and **(d)** a comparison between total column contents from the three OMI $NO_2$ products and Pandora $NO_2$ during May 2016. An average of Pandora 2 h means co-located to valid daily OMI overpasses are overlaid in the spatial plots.

**p.9 l.364: The error on the SCDref is an important source of uncertainty as well, not taken into account in this study. Please include.**

**Response:** we added this sentence "(averaged $NO_2$ VCD obtained from OMI available during KOURS-AQ period is $4.77 \times 10^{15}$ molecules cm$^{-2}$ and standard deviation of $1.33 \times 10^{15}$ molecules cm$^{-2}$, respectively)"

**p.9 l.273: Please clarify how these percentages were obtained. This is unclear. An important uncertainty regarding the AMF is also the uncertainty related to the a priori NO2 profile shape. This is mentioned as well in the conclusion but it is completely ignored in this uncertainty study. Please include.**

**Response:** The $NO_2$ AMF errors were recalculated based on the uncertainties of parameters (AOD, SSA, ALH, and surface reflectance) obtained from previous studies. Relevant references for the uncertainty were also inserted in the revised manuscript (Page 17-18, lines 377−379):

"The σ of AOD, SSA, surface reflectance, and ALH are assumed as 30% (Ahn et al., 2014), 0.04 (Jethva et al., 2014), 0.005+0.05×surface reflectance (EOS Land Validation; https://landval.gsfc.nasa.gov), and 1 km (Fishman et al., 2012), respectively, in this study."

Our paper more focused on the $NO_2$ retrieval from GeoTASO and the investigation of its spatial distribution. Therefore, for now, we considered AOD, SSA, ALH, and surface reflectance in the error analysis section. The authors fully agree with the necessity of AMF error analysis due to the uncertainty related to the a priori $NO_2$ profile shape and preparing it now for the next paper. Some sentences were added to point out the necessity related to $NO_2$ profile shape (Page 19, lines 404−406):

"A priori $NO_2$ profile shape also can be one of factors to cause calculation error for $NO_2$ AMF as

reported in the previous studies (Leitao et al., 2010, Meier et al., 2016). It is necessary to calculate the effect of a priori $NO_2$ profile shape on airborne $NO_2$ AMF error in the future."

- Ahn, C., Torres, O., & Jethva, H. (2014). Assessment of OMI near-UV aerosol optical depth over land. Journal of Geophysical Research: Atmospheres, 119(5), 2457-2473.
- Jethva, H., Torres, O., & Ahn, C. (2014). Global assessment of OMI aerosol single-scattering albedo using ground-based AERONET inversion. Journal of Geophysical Research: Atmospheres, 119(14), 9020-9040.
- EOS Land Validation (https://modis-land.gsfc.nasa.gov/ValStatus.php?ProductID=MOD09)
- Fishman, J., Iraci, L. T., Al-Saadi, J., Chance, K., Chavez, F., Chin, M., ... & Wang, M. (2012). The United States' next generation of atmospheric composition and coastal ecosystem measurements: NASA's Geostationary Coastal and Air Pollution Events (GEO-CAPE) mission. Bulletin of the American Meteorological Society, 93(10), 1547-1566.
- Meier, A. C., Schönhardt, A., Bösch, T., Richter, A., Seyler, A., Ruhtz, T., ... & Burrows, J. P. (2017). High-resolution airborne imaging DOAS measurements of NO 2 above Bucharest during AROMAT. Atmospheric Measurement Techniques, 10(5), 1831-1857.

**p.10 l.283: 7.3% uncertainty on the AMF is really on the optimistic side and not in line with other studies that report around 15%-20%, or larger. Please revise your calculations and/or explain. Just to clarify: it is perfectly fine if you obtain other numbers than in other studies but this should be clarified based on an in-depth analysis. Like mentioned in the previous comment it is unclear where the percentages are coming from, while in other studies these are based on discussed sensitivity tests. Are the numbers provided in Table 6 an average for the flight on 9 June 2016 or what does it represent exactly?**

**Response:** As mentioned above, AMF uncertainties were recalculated based on the uncertainties of input parameters from previous studies. Higher AMF uncertainty (27.8%) has been gained. The numbers in Table 6 are the average for the flight on 9 June 2016. To clarify it, a sentence and the caption for Table 6 have been revised.

Caption for Table 6: "Total errors of $NO_2$ VCD caused by uncertainties in $NO_2$ SCD and $NO_2$ AMF (the average for the flight on 9 June 2016)."

Page 18, line 390: "Table 6 lists the estimated $NO_2$ VCD error on 9 June 2016 for each source based on the error propagation method."

Averages and standard deviations of the parameters (AOD, SSA, aerosol loading height, and surface reflectance) have been added.

Page 18, line 384: "On the flight day, average (standard deviation) values of AOD, SSA, ALH, and surface reflectance were 0.39 (0.10), 0.98 (0.001), 0.27 km (0.10 km), and 0.09 (0.04), respectively."

**Sect. 3.3. all the performed comparisons should report how many measurements/data points are compared (eventually in the plots), in order to assess how statistically relevant the comparisons are.**

**Response:** We revised section 3.3 as follow:

**After modification (P. 21, Line 432-494):**

[revised manuscript text omitted]

In the Busan metropolitan area, the R-value of the $NO_{2,G}$ and $NO_{2,A}$ data had a correlation coefficient greater than 0.78. This reflects the more even horizontal distribution of $NO_2$ in the afternoon, when diffusion from the source areas had taken place. However, for a more accurate comparison, $NO_2$ VCD data should be converted to $NO_2$ MR based on mixing layer height, temperature, and pressure profile data (Kim et al., 2017; Qin et al., 2017; Jeong and Hong, 2021a). However, since the number of pandora and satellite data is limited in this campaign, we had difficulties to validate $NO_2$ retrieved from GeoTASO under various conditions. Since ground-based, airborne and space borne remote sensing measurements has their own advantage and disadvantage, I believe that a comprehensive observation campaign involving all of groud-based, airborne and space borne measurements should be carried out continuously for upcoming new era of geostationary environmental satellite."

**p.10 l.308: Elaborate on this part, eventually with a plot, as many things are unclear! Why are OMI NO2 VCDs only available on 10 June? How did you perform the comparison exactly? Setting a radius is a valid strategy when comparing ground based measurements with airborne or spaceborne data, but not when comparing airborne with satellite. In the latter case you should average the airborne pixels within the larger satellite pixel footprints in order to perform a fair comparison. The time constraint can be kept but could be extended to 1 hour to increase amount of compared measurements. Slope is 0.43: not clear without plot what is under- / overestimating what.**

Response: OMI L2 data (V3 OMNO2) used in this research do not provide some data (30 May to 9 Jun) during the KORUS-AQ period. Therefore, we only have data on 10 June, 2016. Please, see P. 5, Lines 134-136. Also, We revised section 3.3

**Sect. 3.3.1: first sentence should be rewritten as it is unclear. Again not clear: do you compare all pixels within 0.5 km and 30 min with the station measurement or do you average the airborne pixels and do you compare the averaged value? You have hourly station measurements and many flights over the whole campaign period (I think? Much details are missing on flights), but you don't seem to have many data points. Please clarify this. Why not focusing on all possible comparisons between airborne and ground-based measurements instead of restricting to AM and PM in order to improve your statistics. Like it is presented now: not really statistically relevant and without any proper discussion on the non-linear relation between columns and surface concentration, I suggest to leave this section out of the paper.**

Response: We revised section 3.3

**p.11 l.325: Why do you restrict to 5, 9, 10 June? You should clarify the reasons. If you only had flights over this region on these dates that is a proper explanation, but these details have not been provided.**

Response: We thank the reviewer's advice and agree with you for pointing out. We conducted the research focusing on $NO_2$ VCD in the region of interest (megacity and industrial area). The dates (5, 9, 10 June) for calculating $NO_2$ VCDs were determined by performing flight observations in the morning and afternoon because the route are the same only on this three date. To help you understand, we added the content in the manuscript. The flight information is written in P. 5, Lines 123-125

(http://www-air.larc.nasa.gov/missions/korus-aq/docs/KORUS-AQ_Flight_Summaries_ID122.pdf).

**Figure 5: I have a hard time to understand the AMF you obtained. Normally the AMF should be highly correlated with the surface reflectance which isn't really the case here. One (partly) explanation is maybe the rough albedo product used. But this doesn't explain the strong striping in the AMF, especially in the western part. Can you elaborate on this in the manuscript to explain the reasons?**

Response: We apologize for the confusion. We agree with the reviewer's advice. The reason why the

binned AMF (Fig. 5(c)) caused confusion was that there was a technical problem in the plotting process for duplicate pixels when expressing them as a mapping. Therefore, we showed the AMF which is native resolution (250 m) of GeoTASO by modifying Fig. 5(c) and we revised to the contents in the manuscript (P. 13, Line 300).

**Figure 10: I would expect that PANDORA retrievals are higher than the GeoTASO retrievals as the ground-based measurements are more sensitive to the bulk of NO2 close to the surface, but it is the other way around. This could be related to the choices made in the VCD retrievals like mentioned in earlier comments which can strongly affect the absolute values. Are these findings consistent with other GeoTASO-PANDORA comparisons like from the DISCOVER-AQ or LISTOS campaign?**

**Response:** We revised section 3.3

**Minor comments**

**p.3 l.79: SWING has not been operated over Antwerp. The example of Antwerp shown is a simulation. SWING has been operated on a UAV over Romania. It is also better to refer to https://amt.copernicus.org/articles/11/551/2018/amt-11-551-2018.pdf**

**Response:** After revising the manuscript as the comments of reviewers, we have corrected the "Antwerp" mistakes. Please, see P. 3, Lines 67-68.

**p.3 l.81: "regional radiative transfer models" → I think you are referring to regional air quality models here?**

**Response:** "regional radiative transfer models" has been replaced with "regional air quality models" at P. 3, Line 82.

**p.3 l.89: How is transboundary pollution defined here?**

**Response:** Transboundary pollution is pollution that originates from a country but, by crossing the border through pathways of air, is able to cause damage to the atmospheric environment in another country.

♦  Glossary of Environment Statistics, Studies in Methods, Series F, No. 67, United Nations, New York, 1997

**p.4 l.120: Be more clear on how exactly you do the comparison: if an airborne overpass matches the 30 min constraint, do you average spatially all the pixels within 1 km radius? Or do you compare the different individual pixels with the Pandora retrieval? Same comment for p. 5 l. 131.**

**Response:** We revised section 3.3

**p.5 l.143: What is the exact reason of flying so high to measure tropospheric species? At 3 km you**

**would already be well above the PBL. And while indeed the higher flight altitude allows to measure the NO2 in the free troposphere as well, it has the drawback that you are losing some sensitivity towards the NO2 in the PBL and especially at the surface due to larger scattering and absorption probability.**

**Response:** I do not know why NASA observed altitude about 9 km by using GeoTASO. However, I think as follows:

1. Although it is not available to use, but originally GeoTASO also carried out zenith measurement for reference spectrum. For zenith measurement, high altitude flights would be more effective. 2. NASA want to observe a wide area. But it is just my guess. In SIJAQ campaign, NIER have carrying out air borne observation using GCAS measurements on 3~4 km. I hope to use this data to validate GEMS.

**p.7 l.210: Also the PBL plays a role and it's rise in the afternoon + transport/accumulation of emitted NO2 in the PBL during the day. But the latter is already stated later on in this paragraph.**

**Response:** Sorry, we only stated "PBL" in P. 13, Lines 297-298.

**p.8 l.240: I agree with the statement that they can be highly complementary, but it should be clarified to the reader that the relation VCD-surface concentration is highly non-linear. Depending on meteorology it is possible that strong columns are detected while surface concentrations are low. Especially in case of industrial emissions, where the emissions happen at a certain altitude (stacks/chimneys).**

**Response:** We agree with the reviewer's advice and have therefore revised the manuscript as you suggested. We have further explained in P. 24, Lines 480-486.

**p.13 l.376: Link doesn't work**

**Response:** The address of the link is correct. Note the "_" in "~pMENU_NO=125"

**Table 2 and 5: Average mileage is average mileage per car, per day?**

**Response:** We checked average mileage per car per day.

**Technical corrections**

**p.1 l.24: please replace 'data' by 'VCDs'**

**Response:** As you suggested, the sentence was modified on P. 1, Line 24.

**p.2 l.31: you might add 'domestic heating' as well**

**Response:** "domestic heating" has been added to P. 2, Line 32.

**p.2 l.38: cites --> cities**

**Response:** "cites" has been replaced with "cities" on P. 2, Line 39.

**p.2 l.45: what is the point of mentioning UTC here if you don't specify a time?**

**Response:** We have deleted ″(UTC)″ on P. 2, Line 46.

**p.2 l.53: is there a need to repeat here the spaceborne sensors? This was done in the previous paragraph**

**Response:** We deleted the sentence.

**p.3 l.90: replace "from May to June 2016." By "…, organized from May to June 2016."**

**Response:** "from May to June 2016" has been replaced with "organized from May to June 2016" on P. 3, Lines 91-92.

**p.3 l.94: In this study, NO2 VCD retrieval was conducted using solar backscattered radiance observed from GeoTASO over South Korea during the KORUS-AQ campaign --> This sentence is redundant, please remove.**

**Response:** These sentence has been removed from the manuscript.

**p.4 l.106: Amnyeon --> Amnyeon region?**

**Response:** "Amnyeon" has been replaced with "Amnyeon region" on P. 5, Line 129.

**p.4 l.118: O3 → $O_3$**

**Response:** P. 5, Line 413 have been modified.

**p.4 l.122: ". " after Notably**

**Response:** We have added "." to P. 6, Line 149.

**p.5 l.132: "." at end of sentence.**

**Response:** We have added "." to P. 6, Line 159.

**p. 6 l.184: stratospheric NO2 → stratospheric and free tropospheric NO2**

 **Response:** As you suggested, we have modified the manuscript on P. 9, Line 226.

**p. 8 l.219: decrease --> decreases**

**Response:** "decrease" has been replaced with "decreases" on P. 12, Line 292.

**p. 9 l.268: Boersma et al. 2004 --> Boersma et al., 2004**

**Response:** The citation (Boersma et al. 2004) has been corrected to "Boersma et al., 2004"

**Figure 1 caption: please repeat the campaign period here and the number of flights.**

**Response:** Figure 1 has modified following the reviewer's suggestion.

**Figure 3 caption: "." at end of sentence.**

**Response:** We have added "." to P. 8, Line 197.

**Figure 4 caption: remove 'the' in 'the each panel'**

**Response:** We have deleted "the" on P. 11, Line 261.

**Figue 6 caption "into to" --> please correct + what is the meaning of the color-coding of the arrows?**

**Response:** P. 15, Line 327 have been modified.

---

## Author Comment (AC2)

Referee Comment on Atmos. Meas. Tech. Discuss. [preprint], https://doi.org/10.5194/amt-2022-51, in review, 2022.

**Highly resolved mapping of NO2 vertical column densities from GeoTASO measurements over a megacity and industrial area during the KORUS-AQ campaign by Gyo-Hwang Choo at al.**

The article presents airborne and ground-based measurements during the KORUS-AQ field study in South Korea, focussing on NO2 column density measurements above two metropolitan regions and one industrial region in May/June 2016. Observations are performed on several days, in the morning and/or in the afternoon. For the airborne measurements, the GeoTASO instrument is applied. Its specifications, NO2 retrieval details as well as information on the AMF determination are given. Results above the different probed areas are presented and discussed. The probed areas have sparse coverage with trace gas measurements otherwise. Error analysis is included as well as a short comparison with ground-based and OMI observations. Similar to other airborne DOAS sensors, the GeoTASO instrument constitutes a valuable tool for tropospheric trace gas monitoring and mapping, also above areas that are less well accessible. The presented measurements are relevant for a better understanding of the spatial variation of NO2 above South Korean polluted sites that are less well monitored otherwise.

**General Comments**

The overall structure of the article is well understandable. Although there are some typing and grammar mistakes, the text is well readable.

The GeoTASO instrument is introduced and further publications are cited where details of interest can be found. The different assumptions necessary for the conversion from detected slant column densities to meaningful tropospheric vertical column densities are explained.

However, some relevant aspects could be treated with more caution. The respective error analysis could point out more clearly the limitations. While the GeoTASO good spatial resolution is emphasized, this is not really shown, but all data is binned to a 0.01° grid.

The authors rightly consider the comparison between GeoTASO and OMI relevant and mention this in the abstract and conclusions. Therefore, a dedicated figure would certainly support this analysis.

**After consideration of the comments and suggestions below, and after submitting a revised version, I recommend publication of this article in AMT.**

**Response:** First of all, we sincerely apologize for the late submission.

We thank the reviewer's kind comment and advice. The NO2 AMF errors were recalculated based on the uncertainties of parameters (AOD, SSA, ALH, and surface reflectance) obtained from previous studies. Relevant references for the uncertainty were also added in the revised manuscript (P. 17-18, Lines 377–379):

"The  $\sigma$  of AOD, SSA, surface reflectance, and ALH are assumed as 30% (Ahn et al., 2014), 0.04 (Jethva et al., 2014), 0.005+0.05×surface reflectance (EOS Land Validation; https://landval.gsfc.nasa.gov), and 1 km (Fishman et al., 2012), respectively, in this study."

We showed the AMF which is native resolution (250 m) of GeoTASO by modifying Fig. 5(c), and we revised to the contents in the manuscript (P. 13, Line 300). Also, we have revised all the comments

**below.**

- Ahn, C., Torres, O., & Jethva, H. (2014). Assessment of OMI near-UV aerosol optical depth over land. Journal of Geophysical Research: Atmospheres, 119(5), 2457-2473.
- Jethva, H., Torres, O., & Ahn, C. (2014). Global assessment of OMI aerosol single-scattering albedo using ground-based AERONET inversion. Journal of Geophysical Research: Atmospheres, 119(14), 9020-9040.
- EOS Land Validation (https://modis-land.gsfc.nasa.gov/ValStatus.php?ProductID=MOD09)
- Fishman, J., Iraci, L. T., Al-Saadi, J., Chance, K., Chavez, F., Chin, M., ... & Wang, M. (2012). The United States' next generation of atmospheric composition and coastal ecosystem measurements: NASA's Geostationary Coastal and Air Pollution Events (GEO-CAPE) mission. Bulletin of the American Meteorological Society, 93(10), 1547-1566.

**Major Comments**

**-Spatial resolution-**

The article presents observations with different spatial resolution (airborne as compared to satellite), and the usefulness of good spatial resolution is emphasized. Therefore, two aspects should be treated with more care. Firstly, the correct resolution information should be stated. Please update either the figure or correct the caption text of Fig. 1 (the grid for the OMI data is  $0.25^{\circ}$  here). More importantly, the best possible presentation of the GeoTASO spatial resolution should be aimed for. In all figures of the publication, GeoTASO measurements are gridded to a  $0.01^{\circ}$  grid, corresponding to a side length on the order of 1km, while spatial resolution of the instrument is 250m. It is confusing, when 250m resolution is announced and emphasized but never shown. If it is not possible or not aimed at to use the mentioned best resolution, it should be explained in the text why (is the signal-to-noise ratio otherwise too bad? is there another reason?).

**Response**: First, we appreciate the reviewer's comment and apologize for the confusion. We have therefore revised the OMI spatial resolution of the caption in Fig. 1 as you suggested.

Our paper more focused on the NO2 retrieval from GeoTASO and the investigation of its spatial distribution. We did spatial binning after calculating NO2 VCD. Chong et al. (2020) calculated SO2 VCD using data observed in GeoTASO during the KORUS-AQ period, and binning was performed at a resolution of GEMS (7 x 8 km2). As a result of previous studies, it was found that relative random uncertainty decreased as SO2 VCD increased. This random uncertainty can be reduced even when additional spatial binning is performed, which provides a balance between random error and spatial resolution. In conclusion, they found that larger VCDs at 250 m resolutions do not necessarily lead to larger VCDs at 7 km x 8 km resolutions. A similar part of your comments exists in the comments below. Additionally, we showed the AMF by modifying Fig. 5(c) to native resolution to respond to the comments. Please, see P. 10, Lines 249-252.

**After modification (P. 10, Lines 249-252):**

"We showed the finally NO2 VCDs by binning them with  $0.01^{\circ} \times 0.01^{\circ}$  from 250 m spatial resolution. Although the spatial binning NO2 VCDs were compared to those at native resolution, we noted that the spatiotemporal variability was still able to be clearly distinguished from the background at  $0.01^{\circ}$  binning resolution. Chong et al. (2020) showed that larger VCDs at 250 m resolutions do not necessarily lead to larger VCDs at wider resolutions."

• Chong, H., Lee, S., Kim, J., Jeong, U., Li, C., Krotkov, N. A., Nowlan, C. R., ... & Koo, J.-H. (2020). Highresolution mapping of SO2 using airborne observations from the GeoTASO instrument during the KORUS-AQ field study: PCA-based vertical column retrievals. Remote Sensing of Environment, 241, 111725.

**-Treatment of the spectral reflectance-**

Spatial variation of the ground albedo is large, e.g., darker vegetation in parks as compared to paved areas of parking areas, flat roof tops, or similar. The immediate influence on the retrieved slant column of NO2 is substantial, with clear enhancements above the brighter surfaces. The spatial resolution of about 5.6km used for surface characterisation is rather coarse in comparison to the GeoTASO resolution.

Could the intensity of GeoTASO measurements be used to retrieve a pixel-by-pixel ground albedo similar to what is done in Meier et al. 2017 or could another product with better spatial resolution be used? If this is not possible and the coarser spatial resolution shall be retained for the data analysis, the authors should consider a more careful investigation and critical discussion of the albedo treatment and the resulting influence on the error budget. This is also part of the next comment. The uncertainty of the albedo has at least two influencing aspects, (a) the uncertainty in the determination itself in addition to (b) the variability of the albedo within one MODIS ground pixel. Are both aspects considered in the error budget? Is the uncertainty of the combined effect only 20% as stated in the error analysis section?

(Meier, A. C., Schönhardt, A., Bösch, T., Richter, A., Seyler, A., Ruhtz, T., Constantin, D.-E., Shaiganfar, R., Wagner, T., Merlaud, A., Van Roozendael, M., Belegante, L., Nicolae, D., Georgescu, L., and Burrows, J. P.: High-resolution airborne imaging DOAS measurements of NO2 above Bucharest during AROMAT, Atmos. Meas. Tech., 10, 1831–1857, https://doi.org/10.5194/amt-10-1831-2017, 2017.)

**Response:** We agree with reviewer's comments. First of all, as you mentioned, the algorithms for calculating observed surface reflectance using GeoTASO data and surface reflection results such as Meier et al. (2017) were not provided. We tried to use surface reflectance (500 m resolution) of MCD43A3, but there were many pixels that were not produced within the observation area. We wanted to know the spatial distribution of NO2 VCD. Therefore, we used MOD09CMG and MYD09CMG. Because the data exist in all pixels, the surface reflectance.

In this study, the uncertainty according to each input variable was calculated and provided in Section 3.2. In our result, the variable with the largest AMF error was aerosol loading height (26.4%), which is SSA (4.2%), AOD (3.0%), and surface reflectance (2.8%). In addition, the average of the surface reflectance during KORUS-AQ period is 0.055, and there is a study using a fixed ground reflectance of 0.05 to retrieve  $SO_2$  VCD (Chong et al., 2020). As mentioned below, AMF uncertainties were recalculated based on the uncertainties of input parameters from previous studies. Higher AMF uncertainty (27.8%) has been gained. The numbers in Table 6 are the average of the flight on 9 June 2016. To clarify it, a sentence and the caption for Table 6 have been revised. Please, see P. 18, Line 287.

**-Uncertainties-**

The given uncertainties that directly enter the error analysis are not well motivated. Uncertainty values of AOD, SSA, ALH and surface reflectance are assumed and applied (cf. page 9, l. 273), however, no reference or additional information is given. What is the origin of these numbers? The uncertainties seem to be rather small.

**Response**: The NO2 AMF errors were recalculated based on the uncertainties of parameters (AOD, SSA, ALH, and surface reflectance) obtained from previous studies. Relevant references for the uncertainty were also inserted in the revised manuscript (P. 17, Lines 377–379): "The  $\sigma$  of AOD, SSA, surface reflectance, and ALH are assumed as 30% (Ahn et al., 2014), 0.04 (Jethva et al., 2014), 0.005+0.05×surface reflectance (EOS Land Validation; https://landval.gsfc.nasa.gov), and 1 km (Fishman et al., 2012), respectively, in this study."

- Ahn, C., Torres, O., & Jethva, H. (2014). Assessment of OMI near-UV aerosol optical depth over land. Journal of Geophysical Research: Atmospheres, 119(5), 2457-2473.
- Jethva, H., Torres, O., & Ahn, C. (2014). Global assessment of OMI aerosol single-scattering albedo using ground-based AERONET inversion. Journal of Geophysical Research: Atmospheres, 119(14), 9020-9040.
- EOS Land Validation (https://modis-land.gsfc.nasa.gov/ValStatus.php?ProductID=MOD09)
- Fishman, J., Iraci, L. T., Al-Saadi, J., Chance, K., Chavez, F., Chin, M., ... & Wang, M. (2012). The United States' next generation of atmospheric composition and coastal ecosystem measurements: NASA's Geostationary Coastal and Air Pollution Events (GEO-CAPE) mission. Bulletin of the American Meteorological Society, 93(10), 1547-1566.

(For example, as stated above, the surface reflectance within one MODIS 0.05° grid box can vary quite substantially. The resulting uncertainty of the surface reflectance of a GeoTASO ground pixel is given by a combination of the initial uncertainty of the MODIS value, and in addition by this variability of the albedo within one grid box. Is this variability taken into account here?)

**Response:** The resulting uncertainty of the surface reflectance of a GeoTASO ground pixel was not considered in this study. However, this analysis should be carried out in the future. Therefore, the authors added the following sentence in the revised manuscript:

Page 19, Lines 406-408: "Moreover, the resulting uncertainties of input parameters of a GeoTASO ground pixel need to be considered by combining the initial uncertainties of CTM and satellite-based products, and by the variability of the parameters within a grid box. This kind of analysis should be taken into account in further study."

**Also when comparing with typical urban scenarios (Leitao et al., 2010), the influence of aerosol properties (different aerosol types and optical properties) on the AMF would be assumed to be larger than a percent given in ll. 284-286 and Table 6.**

**Response**: The recalculated NO2 AMF errors due to uncertainties in AOD, ALH, and SSA are 3.0%, 26.4%, and 4.2%, respectively. The influence of aerosol properties seems smaller than those in Leitão et al. (2010). It can be explained by aerosol profile (AOD and ALH) and aerosol type (SSA) values on the flight day when error analysis is carried out. The average values of AOD, ALH, and SSA were 0.39, 0.27 km, and 0.98, respectively. Especially, the AOD ranged from 0.15 to 0.68

including low and moderate AOD conditions. As stated in Leitão et al. (2010), the effect of aerosol properties become large in high AOD condition (AOD = 1.05). Therefore, the NO2 AMF errors calculated in this study is smaller than those in the previous study since these were calculated under observation conditions with moderate aerosol loading on 9 June 2016. For better understanding of the readers and according to other comment by the reviewer, we have added the observation conditions on the flight day.

**Furthermore, the results from the spatial variations of the error (ll. 290-305 and Fig.8) yield larger values than stated for the calculated impact (ll. 281-286). This should be reconsidered.**

**Response**: The values in lines 281-286 are averaged AMF errors derived from error propagation method on the flight day. However, values in Figure 8 indicates the percent difference of NO2 AMF on each spatial pixel. Moreover, there are difference in calculation methods. To clarify the difference in the calculation methods, sentences have been added to explain in the revised manuscript. P. 19. Line 409:

$$``AMF_{percent\_diff} = \frac{\partial AMF}{(AMF_{true} + AMF_{new}) \div 2} \times 100$$
(14)''

P. 20, Lines 417-419: In this present study, we additionally investigated the spatial distribution of AMF calculation errors associated with uncertainties in aerosol properties (AOD, SSA, ALH, and SFR). Percent difference of NO2 AMF (AMFpercent\_diff) was calculated on each spatial pixel using Eq. (14)."

In order to understand the situation treated in the study, the field of values (spatial distribution) shown as maps would be helpful, i.e. similar to Fig. 8, four maps giving the applied (unperturbed) values of AOD, SSA, ALH and surface reflectance for an example flight. At least some information of the applied values is needed, such as the average and spread of values used (mean and standard deviations within the measurement area for the above parameters). This would be necessary for the reader to understand the situation.

**Response**: To improve reader's understand, some information (mean and standard deviation values) of the parameters (AOD, SSA, ALH, and surface reflectance) have been provided in the revised text. P. 18, Lines 384-385: "On the flight day, average (standard deviation) values of AOD, SSA, ALH, and surface reflectance were 0.39 (0.10), 0.98 (0.001), 0.27 km (0.10 km), and 0.09 (0.04), respectively."

As correctly stated in the introduction and conclusions, the NO2 vertical profile influences the NO2 AMF. However, this is not explicitly mentioned in the error analysis. For typical urban scenarios, the uncertainty in NO2 profile can add another 10% uncertainty to the AMF (Leitao et al., 2010, Meier et al., 2016). It would be good to take this additional uncertainty into account.

**Response**: Our paper more focused on the NO2 retrieval from GeoTASO and the investigation of its spatial distribution. Therefore, for now, we considered AOD, SSA, ALH, and surface reflectance in the error analysis section. The authors fully agree with the necessity of AMF error analysis due to the uncertainty related to the a priori NO2 profile shape and preparing it now for the next paper. Some sentences were added to point out the necessity related to NO2 profile shape (Page 19, Lines 404-408):

"A priori  $NO_2$  profile shape also can be one of factors to cause calculation error for  $NO_2$  AMF as reported in the previous studies (Leitao et al., 2010, Meier et al., 2016). It is necessary to calculate the effect of a priori  $NO_2$  profile shape on airborne  $NO_2$  AMF error in the future."

**-Resulting error calculation-**

The explanations following eq. (10) especially lines 274-277, are not very well described. Please revise this short part. Was the influence of varying the parameters only determined for positive perturbation as stated (i.e. only for ci + sci and not for ci - sci)? The AMF dependence on the four parameters is non-linear, so that the relative contribution/error source of a negative perturbation could be larger than in positive direction. Both variations (+/- sci) should be investigated. In the analysis of the spatial variations of the error (ll. 290-305), on the other hand, the both-sided influence is rightly investigated.

**Response**: In the error analysis, a negative perturbation was also considered, however, it was not specified in Eq. (13). It has been corrected in the revised manuscript.

**1.** 277 states that ci + sci is the uncertainty of each parameter, which is not correct. This would be only sci. Maybe what is meant is "... the new NO2 AMF simulated using the perturbed input parameters ci + sci (i.e. the original input parameters modified by the uncertainty)".

Response: The authors agree that. The sentence has been modified in the revised manuscript.

**-AMF results-**

Especially when regarding Figure 5 for the Anmyeon region some questions about the AMF calculation arise. While the SCD result from GeoTASO looks reasonable, especially above the two emission plumes, the VCD map shows a stripe of large NO2 values (in latitude direction at about 126.4°E) downwind of the power plant. Looking at the AMF map, the AMF exhibits a sudden low value within this stripe. So the enhanced VCDs seem to be provoked by the low AMF values in this location. In addition, the overall impression of the AMF figure is quite stripy. Hence, the applied AMF values should be checked and potentially corrected if some mistake can be found. It becomes obvious that the AMF influence on the VCD is large, and that possibly the error budget is underestimating the AMF uncertainty. To find the reason, it could be helpful to investigate the spatial distributions of the influencing parameters (AOD, ALH, SSA and SFR).

**Response**: We agree with the reviewer's advice. As you advised, we showed the AOD, ALH, SSA, and surface reflectance used in the AMF calculation in Figure R1. AOD and SSA represent CMAQ

model resolution and SRF represent MODIS resolution. The reason why the binned AMF (Fig. 5(c)) caused confusion was that there was a technical problem in the plotting process for duplicate pixels when expressing them as a mapping. Therefore, we showed the AMF by modifying Fig. 5(c) to native resolution (250 m) of GeoTASO, and revised to the contents in the manuscript (P. 13, Lines 300-305).

---

## Author Response (AR2)

**Review of: Highly resolved mapping of NO2 vertical column densities from GeoTASO** measurements over a megacity and industrial area during the KORUS-AQ campaign**

We thank the reviewer's advice and agree with you for pointing out. In this reviewing process, according your advice, we recalculated NO2 AMF using high resolution surface reflectance data. Furthermore, we selected optimal spectral range for DOAS fitting via sensitivity test. After that we also recalculated NO2 SCD under from 435 nm to 475 nm. As a result, almost all figures were changed, and the output results were also changed (Fig. 5-8 in manuscript).

**Major comments**

-Please extend section 2.1. with campaign information: This comment is not addressed. A link is made to a pdf document but that doesn't contain all relevant info requested. Please add a table to the manuscript with at least the requested info that will improve interpretation of your data and analysis.

**Response**: We added table 1 to section 2.1. Table 1 shows average values from GeoTASO campaign flight observation data such as flight time, flight altitude, SZA, ect. And we added the contents of table 1 to P. 4, line 123 to 126.

| Date                                | 5 Jun       | 9 Jun
AM        | 9 Jun
PM | 10 Jun
AM       | 10 Jun
PM |
|-------------------------------------|-------------|--------------------|-------------|--------------------|--------------|
| ROI                                 | Anmyeon     | Seoul metropolitan |             | Busan metropolitan |              |
| Flight time (LT)                    | 13:11–17:20 | 7:48–12:00         | 13:46–17:52 | 8:02–11:38         | 13:05–15:19  |
| Flight altitude
(km)             | 8.6         | 8.4                | 8.5         | 8.6                | 8.5          |
| Flight speed
(ms -1 ) | 117.0       | 116.2              | 117.6       | 117.2              | 117.1        |
| SZA (°)                             | 39.2        | 36.1               | 45.3        | 35.9               | 33.0         |
| VZA (°)                             | 168.1       | 167.4              | 117.6       | 117.2              | 117.1        |
| AOD                                 | 0.27        | 0.40               | 0.21        | 0.13               | 0.09         |

Table 1. Summary of information on the dates when NO2 VCD was retrieved during the KORUS-AQ period (LT=UTC+9 hr).

**1.**

| SSA                    | 0.966 | 0.980 | 0.949 | 0.981 | 0.968 |
|------------------------|-------|-------|-------|-------|-------|
| Surface
reflectance | 0.07  | 0.09  | 0.09  | 0.06  | 0.06  |
| Cloud fraction         | 0.08  | 0.31  | 0.55  | 0.16  | 0.20  |

-Ref. spectrum: you don't specify the value you are using for the residual amount while this was requested (please check also in TROPOMI or GEMS data what the background value is over your reference area; obviously not possible for the campaign dates but if it is a real background area, it should stay relatively stable). You don't demonstrate that the instrument is stable enough to use a sing ref. spectrum instead of a daily one, like requested. I provided details how you can easily do such a test based on the RMS on the fit.

**Response**: We investigated the time series of total NO2 VCD using TROPOMI offline data on the south of Jeju Island in 2019 and 2020. Fig. R1 shows total NO2 VCD on the south Jeju Island (32.983°N, 126.39°E) in 2019 (black) and 2020 (gray), respectively. In this period, the average of total NO2 VCD is  $4.81 \times 10^{15}$  molecules cm-2 and standard deviation is  $0.43 \times 10^{15}$  molecules cm-2, respectively. During this period, total NO2 VCD is stable.

Figure R1. Total NO2 VCD obtained from TROPOMI in 2019 (black) and 2020 (gray) on South Jeju Island. The red dot represent NO2 VCD obtained from CAMQ.

-Spectral window: You agree that 425-490 nm would be more stable but you don't explain why you took a 425-450 nm window for your analysis. I'm not asking to change to 425-490 nm but I ask to give a proper argumentation for this choice. Moreover, in your answer and subsequent scatter plot you do the analysis for 425-460 nm, while you use the 425-450 nm range in your

**manuscript??? This is not the same...**

**Response**: We recalculated all NO2 SCD in this present study using the radiance from 435 nm to 475 nm and recalculated NO2 AMF at 455 nm. To select optimal wavelength range for DOAS fitting, we carried out sensitivity test with 17 fitting window candidates from 420 nm to 480 nm with the length of the fitting window from 25 nm to 60 nm. We presents this results to Fig. 3 and P. 7, line 197 – 207 in manuscript.

**After modification (P. 7, Line 199 - P. 8, Line 211):**

"The spectral fitting window was selected based on the sensitivity test with 17 fitting window candidates from 420 nm to 480 nm with the length of the fitting window from 25 nm to 60 nm. Spectral fitting residuals and NO2 SCD errors have been investigated for 17 spectral fitting window candidates (Fig. 3). In terms of the residual, when the NO2 fitting window includes a wavelength region less than 430 nm, it tends to have a larger residual compared to the case where it does not. The higher residual can include the more noise signals that cannot be calculated mathematically, which can become an uncertainty for the NO2 SCD retrievals. Therefore, we excluded the fitting window which includes wavelength less than 430 nm for the GeoTASO NO2 retrievals during KORUS-AQ campaign. In case of NO2 SCD error, it was confirmed that the longer the fitting window length, the lower the NO2 SCD error appeared regardless of including the wavelength region less than 430 nm. Therefore, for the stable NO2 SCD retrieval, an appropriate spectral fitting window. To find the optimal fitting window, we set the threshold value based on the results above: residual < 0.001, NO2 SCD error <  $1.4 \times 10^{15}$  molecules cm-2, the length of fitting window > 30 nm. Then, the fitting window of 435–475 nm was finally selected for the GeoTASO NO2 retrievals during KORUS-AQ campaign."

---

## Author Response (AR3)

I'm pleased to accept your revised manuscript "Highly resolved mapping of NO2 vertical column densities from GeoTASO measurements over a megacity and industrial area during the KORUS-AQ campaign" for publication in AMT. While the revisions are minor, there are many of them, so please carefully revise your manuscript before resubmitting it.

* Both reviewers raised several points. Please address each of them in your reply and in the revised manuscript

Again, we sincerely appreciate referees #1 and #2 for their time and critical review of the manuscript which we feel provided an important and neutral perspective on the material presented. As such, the manuscript is much more focused and streamlined. Our point-to-point responses to the reviewer are given below. For clarity, all responses are provided in blue.

* page 1, line 32, remove "domestic heating"

**Response**: As the referee suggested, we removed ″domestic heating″ on P. 1, Line 31.

* page 1, line 35, remove "flash production"

**Response**: As the referee suggested, we deleted ″flash production″ on P. 1, Line 35.

* page 3, line 101: ALH. Throughout the manuscript, you use the term "aerosol loading height" where I guess "aerosol layer height" is meant. When you discuss aerosol treatment, you need to explain your assumptions on the vertical distribution of aerosols. So far, I could not find it in the text but it is important for your AMFs and the uncertainty discussion.

**Response**: We modified ALH to APH (Aerosol Peak Height) in terms of confusion. The APL is assumed to be the height of the highest value in the aerosol extinction coefficient simulated in CMAQ. We added this sentence to P. 9, Line 250-252.

* page 3, line 111: "under Jeju" => ""over Jeju"

**Response**: P. 3, Line 108 was revised as the referee suggested.

* Table 1 and elsewhere: As pointed out by the reviewers, your definition of the VZA is unclear

**and does not seem to be in line with usual definitions.**

**Response:** We had a calculation error. The VZA in Table 1 has been modified.

**\* Figure 2: The fitting window given here is not in agreement with the text**

**Response:** Figure 2 has modified the manuscript on P. 6, Line 175.

**\* Page 7, line 185: As pointed out by the reviewers, information on your reference measurement is given: How many measurements have been averaged? Which slant column has been assumed over this region (you mention the VC from the model). Also, I'm missing a description or formula how you use this SC0 - is that added to all measurements? Please add an explanation in the text. What is the contribution to the uncertainty budget? Please at least mention that in the section on uncertainties.**

**Response:** We appreciate the referee′s comment. We added the following sentences in the revised manuscript as the referee suggested.

"We used the measured radiances at the reference sector to calculate differential slant column density (dSCD) over the whole domain of the GetoTASO measurements. CMAQ calculation over the reference sector (i.e., $6.75 \times 10^{15}$ molecules $cm^{-2}$) was adopted as the reference SCD ($SC_0$), which is added to all dSCD values to convert to the SCD. The reference sector is known as a background area but is occasionally affected by the long-range transport of $NO_2$ from upwind areas. Considering the standard deviation of the OMI measurements accounts for such effects during the measurement period, we estimate the maximum uncertainties of the $SC_0$ can be calculated from this value (i.e., $1.33 \times 10^{15}$ molecules $cm^{-2}$) in addition to the difference of the mean values between the CMAQ and OMI (i.e., $1.98 \times 10^{15}$ molecules $cm^{-2}$). Therefore, our best estimate of the uncertainty of the $SC_0$ is the root of the sum of squares of these values (i.e., $2.38 \times 10^{15}$ molecules $cm^{-2}$)."

**\* page 9, line 248: Ignoring the contribution of the NO2 above the aircraft may be a small error over polluted regions but I frankly don't see why you did that. It would not have been much effort to do the proper calculation.**

**Response:** We appreciate the referee's comment, which we agree with. However, the chemical transport model we used in this study for the AMF calculation (i.e., CMAQ) only simulates the troposphere (surface to 50 hPa), which is why we only consider the $NO_2$ below the aircraft. We added the following sentences in the revised manuscript on page 9, lines 261-264.

*"However, we calculated $NO_2$ VCD↓ by dividing $NO_2$ SCDs by AMF↓ as the CMAQ only simulates the troposphere (surface to 50 hPa). However, as the stratospheric and free tropospheric $NO_2$ ($NO_2$ VCD↑) column densities over megacities and industrial areas are much lower than tropospheric $NO_2$ column densities, (Valks et al., 2011), we assume that the uncertainties in the AMF without considering the upper atmosphere are negligible in this study."*

**\* Table 5 and elsewhere in the manuscript: You talk about "NO2 emission" and "NO2 emission rates" but most of the NOx (NO2 + NO) is emitted into the atmosphere in the form of NO. Therefore, one usually speaks about NOx emissions, not NO2 emissions. In emission inventories, NOx emissions are often given "as NO2", but it still are NOx emissions. Please check and correct in your text where appropriate.**

**Response:** We appreciate the referee's comment, and we revised the manuscript as suggested. ″NO$_2$ emission″ has been replaced with ″NOx emission″.

**\* Figure 7, upper plot: Should there be wind arrows also in the upper plot?**

**Response:** We took a wind arrow to show the wind direction in Figure 7. Due to the resolution of the UM data, only one arrow (126.67°E, 36.985°N) was represented in that domain.

**\* Table 6: As pointed out by both reviewers, the uncertainties given for the reflectance are unrealistically small. Also, the contribution of the NO2 vertical distribution uncertainty, the uncertainty from the assumption on the background column SC0 and the uncertainty from ignoring the NO2 above the aircraft in the AMF calculations should be mentioned here.**

**Response**: The authors modified and added the following sentences in the revised manuscript (P. 17, Line 432-444) to mention the four uncertainties pointed out by both reviewers (1. Unrealistic reflectance error, 2. NO$_2$ vertical distribution uncertainty contribution, 3. An assumption on the background column SC0, 4. Uncertainty from ignoring NO$_2$ above the aircraft in the AMF calculations)

**\* Figure 9: At least in my PDF file, the colour bar does not show exactly the colours used in the figure**

**Response:** We think it looked different because we drew the color with a transparent function. The modified figure shows the transparent function removed.

**\* Figure 9: The change of colour ranges between the figures is confusing. Please use the same range for all sub-plots (-6..+6%)**

**Response:** We thank the referee's comment. However, when we use the same color range, the figure seems harder to understand (please find the below figure). Therefore, we finalized figure 9 as in the revised manuscript rather than the figure below, which are using the same color range.

[Figure]

* **Figure 9 caption: What is the meaning of "20%" after e)?**

**Response:** We have deleted ″20%″ on P. 18, Line 450.

* **Page 18, line 438: SFR is not part of the aerosol properties. Please separate.**

**Response:** ″(AOD, SSA, ALH, and SFR)″ has been replaced with ″(AOD, SSA, and APH), and SFR″ on P. 18, Line 454.

* **Figure 10: Please put GeoTaso data always on the x-Axis**

* **Figure 10: Please always use the same range on x and y-axis, indicate 1:1 line and make the figures quadratic.**

**Response:** We fixed the x-axis with GeoTASO data, and expressed 1:1 line as a solid red line on Fig. 10.

* **Figure 10: Which quantity is shown for the Pandora - tropospheric or total column? If it is total column as I assume, this needs to be made clear in the text. The neglection of the**

stratospheric contribution here and in the AMF calculation may also explain part of the low correlation for small NO2 columns.

**Response:** The $NO_2$ VCD for Pandora is a total $NO_2$ VCD and has been modified in the manuscript (P. 19, Line 481).

**\* Page 21, line 532: 91 => 0.91**

**Response:** P. 21, Line 546 have been modified.

**The paper also needs another round of English proofreading - in parts it is very well written, but some sections are difficult to read.**

**Response:** We corrected the entire manuscript in English.